



# The effects of topography and soil properties on radiocesium concentrations in forest soils in Fukushima, Japan

Misa Yasumiishi[1], Taku Nishimura[2], Jared Aldstadt[1], Sean J. Bennett[1], Thomas Bittner[1]

[1]Department of Geography, State University of New York, the University at Buffalo

[2]Laboratory.of Soil Physics and Soil Hydrology, Graduate School of Agricultural and Life Sciences, The University of Tokyo

*Correspondence to*: Misa Yasumiishi (misayasu@buffalo.edu)

**Abstract.** This research collected forest soil samples from Fukushima, Japan, where the 2011 Fukushima Daiichi Nuclear Power Plant (FDNPP) accident contaminated the land. The purpose of this study was to examine how the local topography influenced the radiocesium (Cs-137) accumulation patterns in soils over the years since the accident. As an analytical method, the general additive model (GAM) was used to determine at what percentages the topographic parameters explain Cs-137

contamination levels down to a depth of 30 cm. For comparison, topographic parameters were extracted from both 1 m and 10 m digital evaluation models (DEMs). The effects of topography were compared with the effects of the soil water content and dry soil bulk density. An additional Tukey's honestly significant difference (HSD) test was conducted to determine the significance of the hillslope aspect and vegetation cover differences on concentration predictions. The results showed that, at this study site, topographic parameters extracted from the 10 m DEM better predicted Cs-137 levels. The models with a single

topographic parameter did not explain Cs-137 levels higher than 30 %. However, combining the parameters improved the explanation percentages. The relative influences of topographic parameters and soil properties were similar throughout the soil depth, showing their subsurface co-functionalities for Cs-137 concentration levels. Tukey's HSD test results showed the inter-effects of topography and vegetation cover differences. The results of this study indicate that the selection of topographic parameters, as well as the chosen methods of their extractions, have implications for physical models assessing radionuclide

contamination levels.



## 1 Background

Radionuclides released from atomic bomb explosions and nuclear accidents have become a research topic for two different purposes: (1) assessing radioactivity levels to protect humans from health risks (Davis, 2016; Fesenko et al., 2007; Tsuda,

2015) and (2) modeling surface soil loss and yields using those radionuclides as environmental tracers (Bennett et al., 2005; Loughran et al., 1987; Lowrance et al., 1988; Mabit et al., 2007; Martz and De Jong, 1987; Pennock et al., 1995; Quine et al., 1997; Ritchie and Ritchie, 1995; Wallbrink et al., 2002; Walling et al., 2007; Xinbao et al., 1990). One of the most commonly researched radionuclides is radiocesium (cesium-137; Cs-137). Cs-137 is a byproduct of nuclear fission, which is a mechanism for generating energy, and it does not naturally exist in the environment (Amaral et al., 1998; IAEA, 2015; Tsoulfanidis, 2012).

Atomic bomb explosions began emitting Cs-137 into the atmosphere in the late 1940s (Mahara, 1993), and emissions were highest from the late 1950s through the early 1960s, when atomic bomb testing was most active during the Cold War (Ritchie and McHenry, 1990; The Arms Control Association, 2019).

Cs-137 emits gamma-ray radiation, and this radiation can be harmful to organisms – including humans – when they are exposed

to or inhale Cs-137 or intake Cs-137 via contaminated food or water at a high dosage (EPA, 2012; IAEA, 2019; Wrixon, 2004). The negative effects of radiation exposure on humans range from minor physiological symptoms to mortality (IAEA, 2001; IAEA, 2020; Scott, 2004). Biological consequences in plants and animals have been reported after the Chernobyl Nuclear Power Plant accident in 1986 (Barker et al., 1996; Kovalchuk et al., 2000; Møller and Mousseau, 2006).

Once released, Cs-137 takes two pathways in the environment. It may be dissolved in water (Iwagami et al., 2015; Osawa et al., 2018; Sakuma et al., 2018; Tsuji et al., 2016) or adsorbed into soil particles. Cs-137 is positively charged and has an affinity for clay minerals, which are negatively charged because of their ion composition in a crystal structure. Cs-137 can, thus, be fixed in clay mineral structures by exchanging positions with other positive elements, such as magnesium (Claverie et al., 2019; Fan et al., 2014; Murota et al., 2016; Nagao, 2016; Nakao et al., 2008; Ohnuki and Kozai, 2013; Park et al., 2019; Ritchie

and Ritchie, 1995).

When used as a tracer for geomorphological analysis, Cs-137 becomes a signal indicating where soils—particularly clay soils—accumulate or are lost. Whether Cs-137 moves by being dissolved in water or with soil particles, topography plays an important role in the process. It determines how precipitation flows on the ground surface and migrate subsurface and, hence,

determines the directions of Cs-137's movement on the surface and its concentration levels in soils (Komissarov and Ogura, 2017; Martin, 2000; Roering et al., 2001; Roering et al., 1999; Schimmack et al., 1994; Schimmack et al., 1989; Teramage et al., 2014).



In Fukushima, Japan, the Fukushima Daiichi Nuclear Power Plant (FDNPP) underwent hydrogen explosions in March 2011.
The accident was caused by a power outage at the plant because, following a strong earthquake, a tsunami damaged the plant's
power generators (IAEA, 2015; Mahaffey, 2014). The environment in nearby towns was contaminated by the radioactive
plume released from the plant. Since the accident, numerous field research activities have taken place in the region to assess
contamination levels and devise remediation strategies. One particular interest among researchers has been examining the Cs-
137 accumulation differences between Fukushima and the region affected by the Chernobyl Nuclear Power Plant (CNPP)
accident in 1986. The CNPP and the FDNPP accidents are the only two nuclear power plant accidents categorized as Level 7
– the highest level on the International Nuclear and Radiological Event Scale (INES) (IAEA, 2015; IAEA and INES, 2008;
Mahaffey, 2014). However, the climates and topographies of the Fukushima and Chernobyl regions differ (Table 1), and
researchers have suspected that Cs-137 may behave differently in both areas.

Table 1. A snapshot of climates, topographies, and soil formations between the FDNPP accident–affected area and the CNPP
accident–affected area.

|  | FDNPP accident–affected area | CNPP accident–affected area |
| --- | --- | --- |
| Climate | Cfa (humid climate; Köppen climate classification system). | Dfb (Warm-summer humid continental climate; Köppen climate classification system). |
|  | Average annual precipitation: 1,361.6 mm (Japan Meteorological Agency, 2019). | Average annual precipitation: 621 mm (Climate-Data.Org, 2019). |
| Topography | Mountainous. Forests are covered with deciduous and evergreen trees. | Steppes and plateaus. |
| Soil formation | Volcanic activities and weathering, deposition. | Glacial activities and weathering, deposition. |

## 1.1 Literature review and aims of this study

The authors reviewed 30 articles published after the FDNPP accident, all of which conducted soil sampling in the Fukushima
region (Table 2). These articles described sampling timing and locations, land-use types, sample collection methods, and
sample depth. However, only two projects incorporated topographic parameters into their assessments because early research
concerns included contamination in urban areas and rice paddies since rice is the staple food in Eastern Asian countries. The
previous literature, assessing the radioactive contamination in Eastern Europe following the CNPP accident, has examined
contamination levels by taking topography into account (e.g., elevation, slope, and aspect) (Korobova and Romanov, 2009;
Korobova and Romanov, 2011; Korobova et al., 2019; Linnik et al., 2020).



Based on this background and literature review, the current study examined how well topographic parameters explain and predict Cs-137 concentration levels. Since about 70 % of the Fukushima region is forested (Japan Forest Agency, 2017), this study used forest soils as a research subject. For analysis, four steps were incorporated.

1)  Test 1: Conducting a descriptive analysis of Cs-137 concentration patterns on a simple, representative hillslope.
2)  Test 2: Examining the explanation power of multiple topographic parameters on Cs-137 accumulation levels.
3)  Test 3: Examining the effect of vegetation cover and locational grouping on Cs-137 level predictions if Test 2 did not return significant results.
4)  Test 4: Examining the applicability of the results of Test 2 in a basin-wide spatial Cs-137 prediction.

This case study sought to provide a further understanding of Cs-137's behavior in forest soils and to contribute to future response and remediation strategies for radioactive contamination.

**Table 2. Soil sampling projects conducted in the Fukushima region following the 2011 FDNPP accident.**

| Article # | Authors | Publication year | Sampling year | Land use type | Soil collection methods | Deepest sample depth | Reporting unit | Topographic parameters |
|---|---|---|---|---|---|---|---|---|
| 1 | Shiozawa et al. | 2011 | 2011 | Rice paddies | Scoop and cylinder | 15 cm | Bq kg$^{-1}$ Bq m$^{-2}$ | N/A |
| 2 | Tagami et al. | 2011 | 2011 | Flower garden | Scoop | 12 cm | Bq kg$^{-1}$ | N/A |
| 3 | Tanaka et al. | 2012 | 2011 | Field, orchard | Stainless steel pipe | 30 cm | Bq kg$^{-1}$ Bq m$^{-2}$ | N/A |
| 4 | Fujiwara et al. | 2012 | 2011 | Forest, rice paddy, urban | House-made soil sampler | 30 cm | Bq kg$^{-1}$ Bq m$^{-2}$ | N/A |
| 5 | Koarashi et al. | 2012 | 2011 | Croplands, grasslands, pastures | Core sampler | 20 cm | Bq kg$^{-1}$ Bq m$^{-2}$ | Altitude |
| 6 | Kato et al. | 2012 | 2011 | Home garden at a residence | Scraper plate | 30 cm | Bq kg$^{-1}$ Bg m$^{-2}$ | N/A |
| 7 | Ohno et al. | 2012 | 2011 | Wheat fields, rice paddies, orchards, and forestland | A stainless steel core sampler | 20 cm | Bq kg$^{-1}$ Bq m$^{-2}$ | N/A |
| 8 | Endo et al. | 2012 | 2011 | Uncultivated lands, such as shrubs, school playgrounds, flowerbeds, and a sandbox (in a park) | N/A | 10 cm | Bq g$^{-1}$ Bq m$^{-2}$ | N/A |
| 9 | Yamamoto et al. | 2012 | 2011 | Roadside, school playgrounds | Stainless steel pipe, soil sampler | 30 cm | Bq kg$^{-1}$ Bq m$^{-2}$ | N/A |



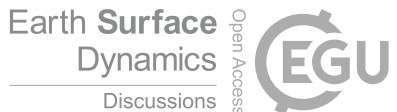

| | | | | and paddy or dry fields | | | | |
|---|---|---|---|---|---|---|---|---|
| 10 | Taira et al. | 2012 | 2011 | Undisturbed surface soils | Core sampling | 10 cm | Bq kg$^{-1}$ Bq m$^{-2}$ | N/A |
| 11 | Zheng et al. | 2012 | 2011 | Research center ground, public park, garden, forest | N/A | 13 cm | mBq g$^{-1}$ | N/A |
| 12 | Endo et al. | 2013 | 2011 | Rice paddies, uncultivated land | N/A | 30 cm | Bq g$^{-1}$ Bq m$^{-2}$ | N/A |
| 13 | Nakanishi et al. | 2013 | 2011 | General farming field, vegetable field, wheat field, paddy soil field for rice | N/A | N/A | Bq kg$^{-1}$ | N/A |
| 14 | Matsunaga et al. | 2013 | 2011 | Croplands, grasslands, forests | Core sampler | 25 cm | Bq m$^{-2}$ | N/A |
| 15 | Tanaka et al. | 2013 | 2012 | Paddy field | Plastic corer, stainless steel pipe equipped with an inner plastic corer | 30 cm | Bq g$^{-1}$ Bq m$^{-2}$ | N/A |
| 16 | Saito et al. | 2014 | 2011 | N/A | House-made soil sampler | 5 cm | Bq m$^{-2}$ | N/A |
| 17 | Takata et al. | 2014 | 2011–2012 | Upland, paddy fields, orchard, meadow | Hand sampler | 15 cm | Bq kg$^{-1}$ | N/A |
| 18 | Sakai et al. | 2014 | 2011–2012 | Rice paddies | Core sampler | 20 cm | Bq kg$^{-1}$ | N/A |
| 19 | Nakanishi et al. | 2014 | 2011–2012 | Forest | N/A | 10 cm | Bq kg$^{-1}$ Bq m-2 | N/A |
| 20 | Fujii et al. | 2014 | 2011–2012 | Forest | Core sampler | 20 cm | Bq kg$^{-1}$ Bq m$^{-2}$ | N/A |
| 21 | Yoshikawa et al. | 2014 | 2012 | Paddy field | Soil sampler | 15 cm | Bq kg$^{-1}$ | N/A |
| 22 | Takahashi et al. | 2015 | 2011–2012 | Forests, pasture, meadow, farmland, tobacco field, rice paddy | Scraper plate | 10 cm | Bq kg$^{-1}$ Bq m$^{-2}$ | Altitude |
| 23 | Maekawa et al. | 2015 | 2011–2012 | N/A | Stainless steel tube | 15 cm | Bq kg$^{-1}$ Bq m$^{-2}$ | N/A |
| 24 | Matsuda et al. | 2015 | 2011–2012 | N/A | Scraper plate | 8 cm | Bq kg$^{-1}$ Bq m$^{-2}$ | N/A |



| 25 | Saito et al. | 2015 | 2011 | Fields with little vegetation (farm fields were avoided) | Referred to Onda et al. (2015) | 5 cm | Bq m$^{-2}$ | N/A |
|----|----|----|----|----|----|----|----|----|
| 26 | Lepage et al. | 2015 | 2013 | Paddy fields | Augur | 20 cm | Bq kg$^{-1}$ Bq m$^{-2}$ | N/A |
| 27 | Onda et al. | 2015 | 2011 | Forest floor, grassland, and paddy field | Plastic container, core sampler | 5 cm | Bq kg$^{-1}$ Bq m$^{-2}$ | N/A |
| 28 | Yang et al. | 2016 | 2011–2014 | Rice paddies | Referred to Onda et al. (2015) | 30 cm | Bq kg$^{-1}$ Bq m$^{-2}$ | N/A |
| 29 | Ayabe et al. | 2017 | 2013–2015 | Secondary forests | N/A | 5 cm | Bq kg$^{-1}$ Bq m$^{-2}$ | N/A |
| 30 | Wakai et al. | 2019 | 2014 | Roadside, paddy, upland, canal ditch, mountain | Soil sampling scoop | 5 cm | Bq kg$^{-1}$ | N/A |

## 2 Study site description

### 2.1 Topography

The soil sampling site was in a forest in Iitate Village, Fukushima, Japan, which is located about 35 km northwest of the FDNPP (Fig. 1). Hydrogen explosions at the FDNPP occurred multiple times, and the radioactive plume was carried in different directions and at various distances with the wind (Hashimoto et al., 2012; IAEA, 2015; MEXT, 2011). Figure 2 shows the timing and main locations of the Cs-137 fallout (IAEA, 2015). The study site was under the plume released on the morning of 15 March 2011 from FDNPP Unit 2. The plume initially moved toward the southwest and then, following a change in wind direction, toward the northwest, resulting in wet deposition across northwestern Fukushima and other prefectures (IAEA, 2015).



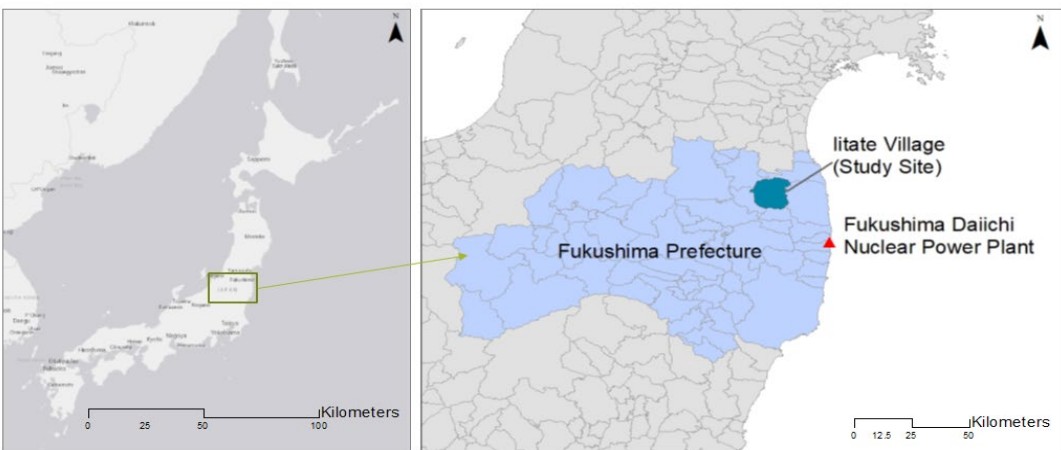

**Figure 1. Locations of the FDNPP and the study site (Basemap: ESRI, HERE, Garmin, © OpenStreetMap contributors. Distributed under a Creative Commons BY-SA License.; Iitate Village, Fukushima, Japan (Japan municipal border map: © ESRI Japan).**

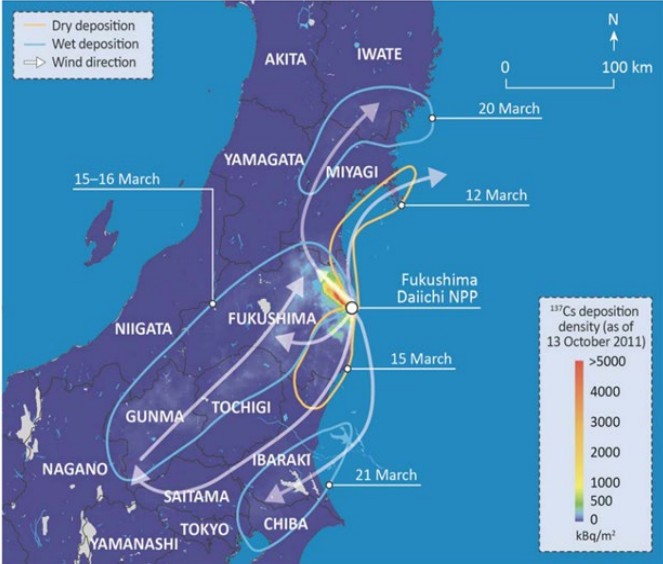

**Figure 2. Timing and locations of the main Cs-137 deposition events following the explosions at the FDNPP (Map: IAEA [2015]).**

When driving into the study site region, visitors see hills and mountains covered with forests. Winding, narrow roads connect the town center, residential areas, and farmlands scattered across the lowlands (Fig. 3). The geology of the Fukushima region is mostly composed of Paleozoic metamorphic rocks and Paleo-Mesozoic igneous rocks (Forest Management Center, 2017). The mountains in the region are low-rising mountains with micro-topographies, where narrow – and sometimes steep – streams cut through the hillslopes.



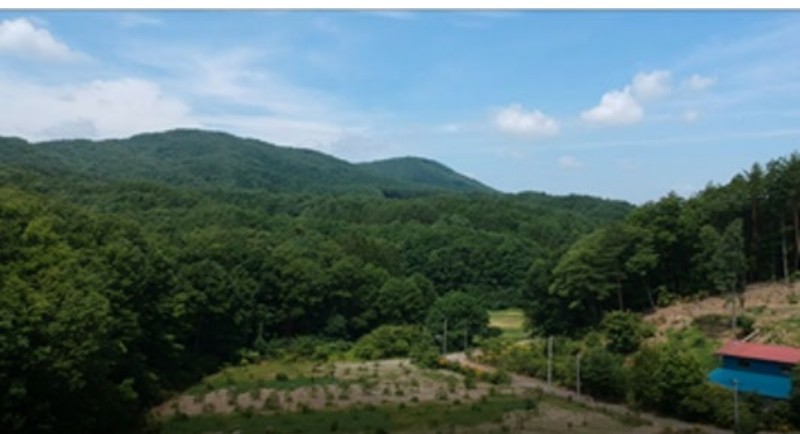


**Figure 3. Aerial views of the study site, facing northwest. The image shows the road stretching toward the village center (shot in 2018).**

These mountains are covered in deciduous and conifer trees, and litter accumulates on the forest floor. During spring and summer, canopies cover the hills and the sky is typically not visible from the ground. However, at the particular study site in this research, the canopies are not thick at the tops of the hills. Standing on the highest ridge, hikers have a view of nearby mountains rolling toward the southeast; however, the FDNPP is not visible from this ridge.

These forests are not all "natural" or "native." The Japanese government began tree-planting and land management projects in
the late seventeenth century to mitigate the overharvesting of lumber and land degradation due to the country's increasing population. Two types of trees that the government recommended for planting in northern Japan were cedar and Japanese cypress (Ministry of Agriculture Forestry and Fisheries, 2013). Whether the land-use history of the forests affects radionuclide behavior is an important question. However, since the radionuclide fallout happened recently, and since no major forestry work (e.g., new planting) has been conducted in the area after the accident, land-use history was not considered in the following
analysis.

## 2.2 Background contamination levels

Understanding Cs-137 contamination requires the determination of background contamination levels. Background Cs-137 contamination data before the FDNPP accident were unavailable for the forests sampled in this study. According to previous literature, in Japan, Cs-137 levels have varied from 15 Bq kg$^{-1}$ or lower to 100 Bq kg$^{-1}$, although the 100 Bq kg$^{-1}$ measurement
was an outlier (Table 3). Conservatively, 100 Bq kg$^{-1}$ was used as the background contamination level in the following analysis.



**Table 3. Background levels of Cs-137 in soils in Japan before the FDNPP accident (the highest concentration at a measured soil depth).**

| Location and approx. distance from FDNPP | Sample year | Highest concentration (approx.) | Measured depth (approx.) |
|---|---|---|---|
| Ibaragi (180 km southwest of FDNPP; Yamaguchi, et al., 2012) | 1996 | 50 Bq kg$^{-1}$ (forest) | 10 cm |
| Sea of Japan side (Komamura et al., 2006) | 1959–1978 | 100 Bq kg$^{-1}$ (rice paddy) | NA |
| Aomori (350 km north of Fukushima; Tsukada et al., 2002) | 1996-1997 | 15 Bq kg$^{-1}$ (paddy soil) | 5–20 cm |
| Fukushima City, Fukushima (60 km northwest of Fukushima; MEXT, 2006) | 2005 | 21 Bq kg$^{-1}$ | 0–5 cm |

**3 Soil sampling**

**3.1 Field sampling**

Soil samples were collected in 2016, 2017, and 2018 during summer (Fig. 4). In 2016, samples were collected at 21 locations to cover accessible hillslope areas from the lowlands in a circular pattern. In 2017, only a limited number of samples were collected at six locations. The sampling was partly for confirmatory purposes at locations that had Cs-137 or soil-type

anomalies in 2016. In 2018, the sampling campaign focused on the southwest slope, and the sample locations covered the entire slope. Additional samples on the eastern side were also collected in 2018. However, most of the sampling locations were on the southwest slope because of the accessible walk routes from the hill bottom to the ridges. There were 46 total sampling locations. At eight locations, multiple-year samples were collected, and those measurements were averaged in this article. The total number of samples, including those multiple-year samples, was 58.


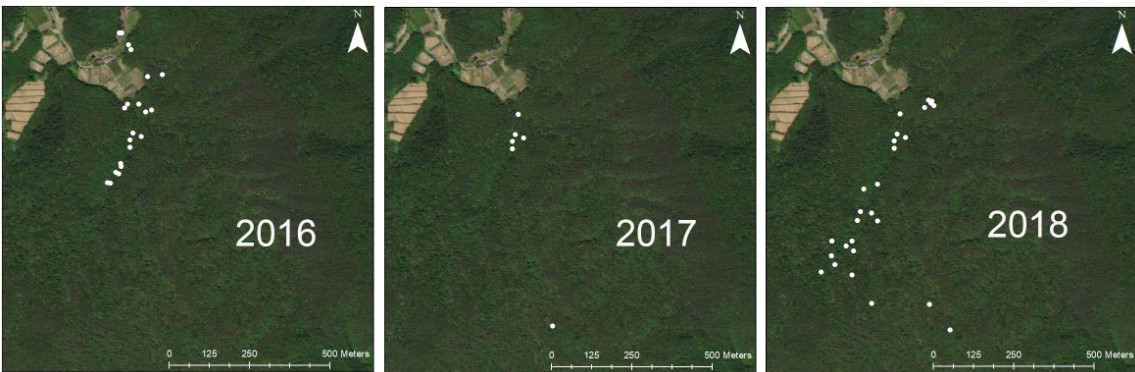

**Figure 4. Sampling location developments through three summers (Basemap: Esri, DigitalGlobe. GeoEye, Earthstar Geographics, CNES/Airbus DS, USDA, USGS, AeroGrid, IGN, and the GIS user community).**



The images in Fig. 5a–Fig. 5f show the study site's topography and sampling locations (yellow dots). The elevation change
from the lowest to the highest sampling point was approximately 140 m (Fig. 5b). The 5 m interval contour lines in Fig. 5c
were extracted from a 1 m digital evaluation model (DEM) using ArcMap 10.7.1 (ESRI Inc., 2019). The satellite imagery
credit will be abbreviated as "ESRI" hereafter. The largest basin border at the study site (Fig. 5d) was extracted using the
System for Automated Geoscientific Analyses (SAGA GIS; Conrad et al., 2015). The area of the largest basin is 0.56 km².
The flow directions (Fig. 5e) were determined by the "D-infinity Flow Direction" tool of Terrain Analysis Using Digital
Elevation Models (TauDEM; Tarboton, 1997). As Fig. 5f – an image taken in winter – shows, the ground is exposed in winter
except for areas with evergreen trees.

A soil sampler (diameter: 5 cm; length: 30 cm) from Daiki Rika Kogyo Co., Ltd., Japan, was used for soil sample collection.
The tube was made of metal and contained a replaceable plastic liner. The sampler was pushed into the ground with a hammer.
Once the sampler was fully inserted, it was pulled out and the plastic liner containing the soil was removed, sealed, and brought
back to a laboratory at the University of Tokyo.



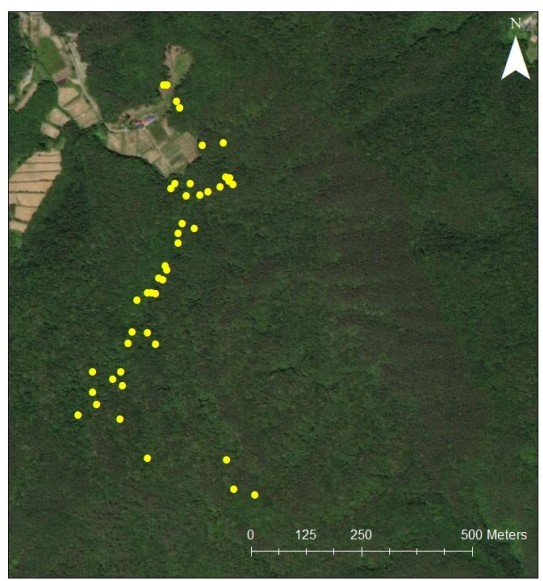

**(5a) Aerial image of the study site (Basemap: ESRI).**

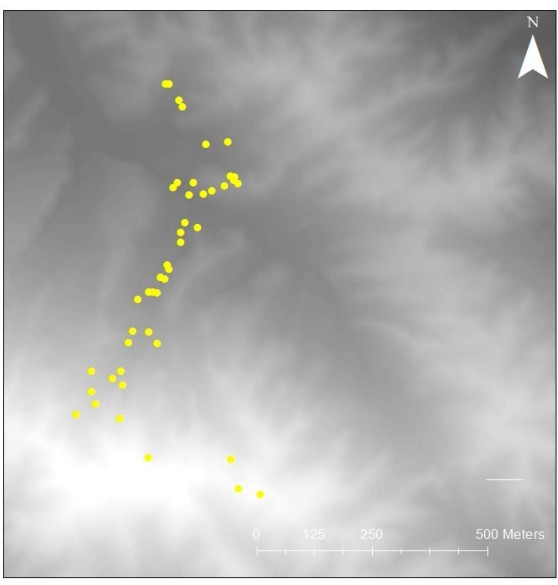

**(5b) Elevation (m) profile of the study site.**

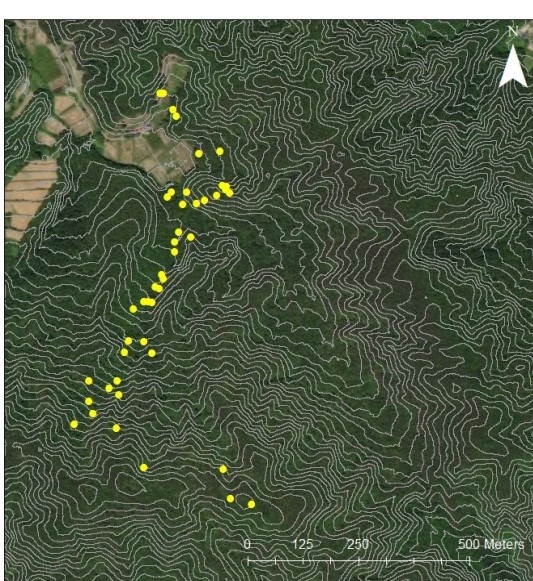

**(5c) Contour lines of the study site (5 m interval) (Basemap: ESRI).**

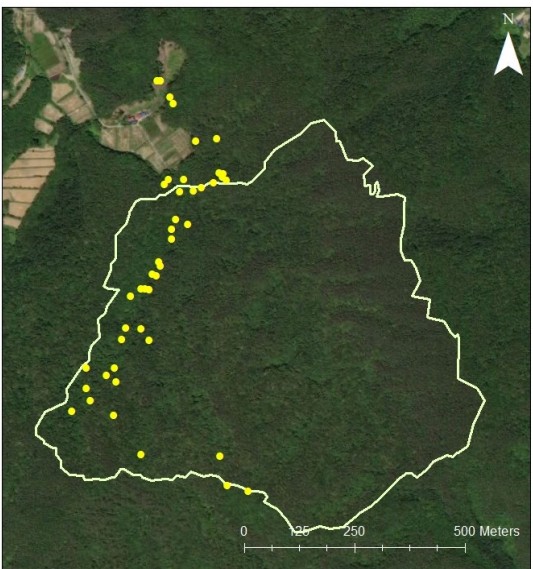

**(5d) A large basin enclosing the longest slope (Basemap: ESRI).**





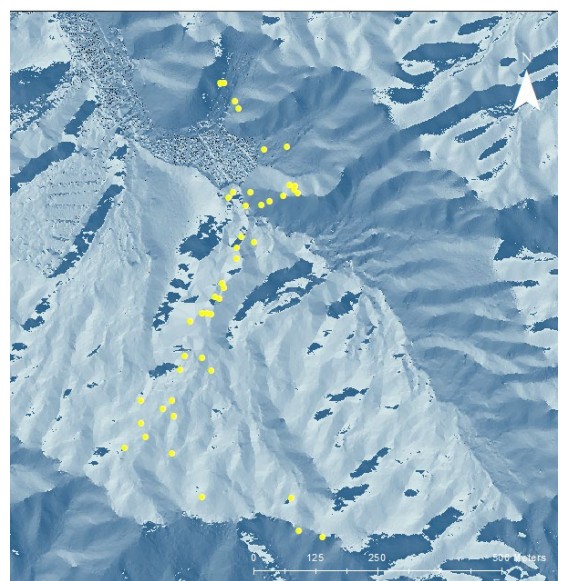

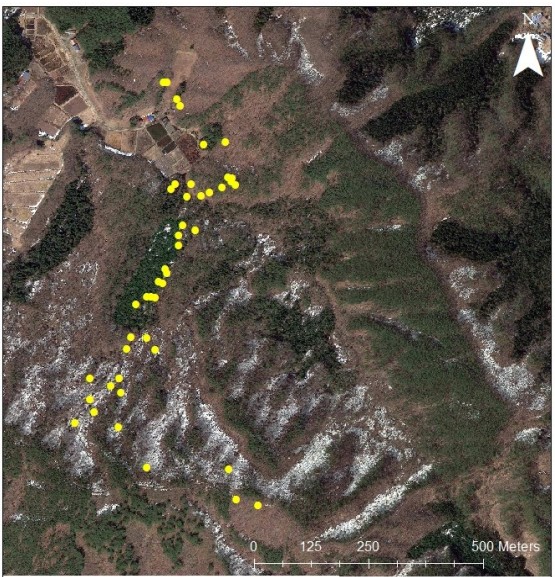

**(5e) Flow direction differences of the study site. (Darker colors flow between 135 and 315 degrees. Lighter colors flow between 315 and 135 degrees.)**

**(5f) Deciduous and evergreen vegetation at the study site as of 9 January 2017 (Map data: Google, Maxar Technologies).**

**Figure 5. Study site topography and aerial images.**


## 3.2 Radioactivity Measurement

In the laboratory, each plastic tube was cut open with a knife, photographed, divided into 2.0 cm-thick disks (from the surface to 20.0 cm deep), and 2.5 cm thick disks (20.0–30.0 cm deep). In this article, the total value in an entire tube is called a "core total." When a sample contained visible pieces of plant roots or rocks, these pieces were removed. Each disk sample was

placed on a plate, weighed, and dried in an oven for about 24 hours at 105°C. The dried sample was reweighed, placed in a mortar, and ground into fine pieces. When the dried sample contained soil particles larger than 2.0 mm (that is, larger than the sand category), these particles were removed using a sieve. Other than these removals, no additional sorting or washing was applied.

Processed samples were stored in polyethylene vials and sent to the Isotope Facility for Agricultural Education and Research, Graduate School of the University of Tokyo, for isotope analysis. Radioactivity levels were measured with a NaI(Tl) scintillation automatic gamma counter (2480 WIZARD2 gamma counter, PerkinElmer Inc., Waltham, MA, US), which was equipped with a well-type NaI(Tl) crystal (diameter: 3 in; length: 3 in) and covered with a 75 mm thick lead shield. Energy calibrations were performed using the 662 keV energy peak of gamma rays from Cs-137. It should be noted that "keV" is the

abbreviation for "kiloelectron volts," a unit of energy in diagnostic radiography and nuclear medicine equivalent to the kinetic



energy gained by an electron falling through a potential of 1 volt. For radiocesium, the detection limit was approximately 0.5
Bq. After each measurement, the radiation was separated into radiation emitted by Cs-137 and Cs-134 using the abundance
ratio of Cs-137 to Cs-134 at the time of sampling. That ratio was obtained from the physical decay rates of the isotopes and
the elapsed time from the accident to sampling, assuming that the ratio at the time of the FDNPP accident was 1:1 (Nobori et
al., 2013; Tanoi et al., 2019). Although a gamma-ray spectrometer provides more precise Cs-137 measurement data, lower-
resolution NaI with effective algorithms can be quite effective and, thus, is preferred for its ruggedness, shorter time required
for evaluation, and cost of operation (Burr and Hamada, 2009; Stinnett and Sullivan, 2013). The United States Environmental
Protection Agency (US EPA) allows this NaI method for gamma-ray measurement during radioactive incident responses (EPA,
2012).

**3.3 Soil property measurement**

During soil processing in the laboratory, soil water content (%) and soil dry bulk density (g cm$^{-3}$) were calculated for each
disk. Table 4 displays the depth layer averages of water content (%) and dry bulk density (g cm$^{-3}$), average standard deviations,
and the average coefficient of variations (COVs: standard deviation/mean). In the top layer, the average water content
percentage was above 100 % because some samples were very moist. Dry bulk density was low (<1.0 g cm$^{-3}$) throughout the
depth, although it increased as the soil depth increased. The standard deviations of water content in the uppermost soil layers
were large (>0.50) and decreased in deeper soils– an opposite trend to the standard deviations of bulk density. The COVs of
water content and bulk density were approximately the same in the top 4.0 cm. Both COVs decreased at the 6.0–10.0 cm depth
then, once again increased toward the mid-depth of 16.0–20.0 cm. Figure 6 shows the measured value ranges, mean, median,
and COVs of water content and dry soil bulk density. Water content percentages showed a wider variation (COV) than soil
bulk density. The average bulk density of all the samples was 0.44 g cm$^{-3}$.

Previous studies have indicated that a soil's texture affects the amount of adsorbed Cs-137 per unit mass of soil particles and,
thus, the accumulation patterns of Cs-137 in soils (Bennett et al., 2005; Giannakopoulou et al., 2007; Korobova et al., 1998;
Walling and Quine, 1992). Thus, a portion of the soil samples was selected for soil texture testing (50 % of the sampling
locations). Only some of the samples were tested for texture because of time and human resource constraints during the limited
lengths of the first author's stays in Japan. The samples chosen for texture testing were selected so that they covered various
soil types and colors and so that their locations were not concentrated at a certain elevation.





**Table 4. Soil properties of the samples in this article by depths.**

| Depth (cm) | Average water content (%) | Average bulk density (g cm⁻³) | Average standard deviation | | Average COV | |
|---|---|---|---|---|---|---|
| | | | Water content (%) | Bulk density (g cm⁻³) | Water content (%) | Bulk density (g cm⁻³) |
| 0.0–2.0 | 1.22 | 0.24 | 0.72 | 0.14 | 0.59 | 0.59 |
| 2.0–4.0 | 0.99 | 0.28 | 0.56 | 0.15 | 0.57 | 0.54 |
| 4.0–6.0 | 0.90 | 0.32 | 0.52 | 0.13 | 0.58 | 0.40 |
| 6.0–8.0 | 0.79 | 0.38 | 0.42 | 0.15 | 0.53 | 0.40 |
| 8.0–10.0 | 0.69 | 0.45 | 0.31 | 0.17 | 0.45 | 0.39 |
| 10.0–12.0 | 0.72 | 0.47 | 0.45 | 0.20 | 0.62 | 0.42 |
| 12.0–14.0 | 0.63 | 0.48 | 0.34 | 0.19 | 0.54 | 0.39 |
| 14.0–16.0 | 0.66 | 0.50 | 0.41 | 0.20 | 0.62 | 0.41 |
| 16.0–18.0 | 0.59 | 0.52 | 0.39 | 0.23 | 0.65 | 0.44 |
| 18.0–20.0 | 0.56 | 0.57 | 0.36 | 0.27 | 0.65 | 0.48 |
| 20.0–22.5 | 0.56 | 0.54 | 0.34 | 0.23 | 0.61 | 0.43 |
| 22.5–25.0 | 0.53 | 0.54 | 0.33 | 0.23 | 0.63 | 0.43 |
| 25.0–27.5 | 0.53 | 0.51 | 0.33 | 0.22 | 0.63 | 0.43 |
| 27.5–30.0 | 0.48 | 0.51 | 0.27 | 0.21 | 0.55 | 0.42 |

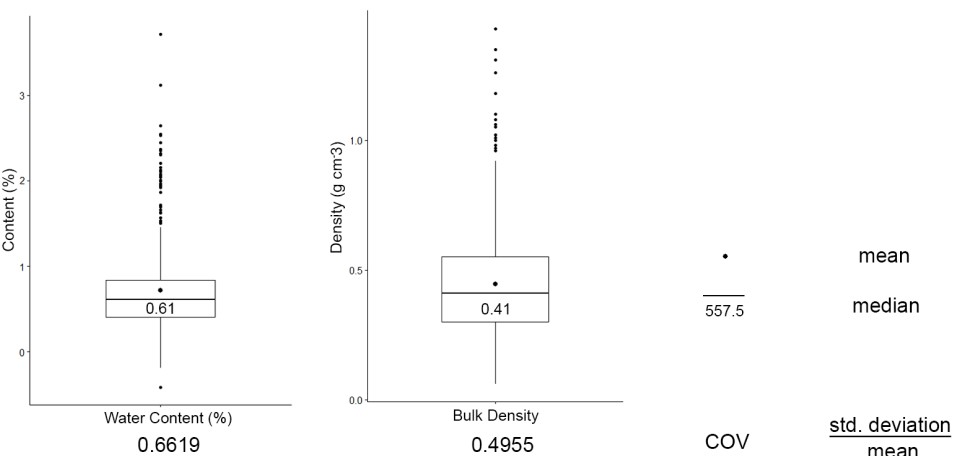

**Figure 6. Measurement distributions of water content (%) and dry bulk density (g cm⁻³).**

On average, the tested soils contained more than 50 % sand, and most of the samples could be categorized as sand, loamy sand, sandy loam, and loam (Fig. 7).



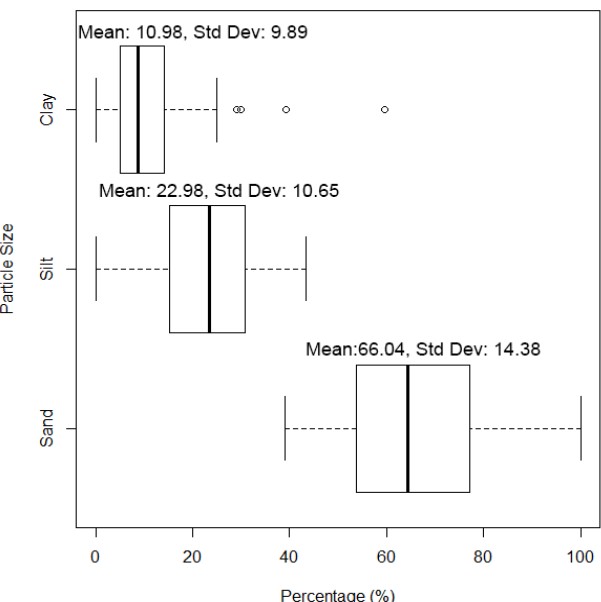

**Figure 7. Percentages of clay, silt, and sand particles in the selected soil samples.**

## 4 Data

### 4.1 DEMs

In this study, the topographic effects were examined via two DEMs with different spatial resolutions: 1 m and 10 m. Certain topographic features depend on spatial resolutions, and using multiple DEMs enables researchers to identify dependency (Gallant and Wilson, 1996; Kim and Lee, 2004; Moore et al., 1993). The 1 m resolution DEM was provided by Forest Research and Management, Japan. Its original datum was GCS JGD 2011 (Zone 9), and the data collection year was 2012. The 10 m resolution DEM was downloaded from the website of the Geospatial Information Authority of Japan (https://www.gsi.go.jp/kiban/). Its file date was 1 October 2016. The 10 m DEM was generated by the Japanese government, and the original datum was GCS JGD 2000. The coordinate projection of all GIS files, including DEMs, was set to UTM 54N in this article.

### 4.2 Topographic parameters

The following list shows the original parameters used in this analysis as well as the guiding assumption behind the parameter selection.

1) **Elevation:** Soil particles, by which Cs-137 is adsorbed, move downward on a sloped surface. Thus, Cs-137 levels are higher in downslope areas (Martin, 2000; Roering et al., 1999).





2)  **Slope degrees, upslope distance to a basin edge:** Relating to elevation, at a certain point on a hillslope, Cs-137
levels are controlled by slope degrees and the length of the hillslope above that point (Komissarov and Ogura, 2017;
       Roering et al., 2001; Roering et al., 1999).

3)  **Distance from the hill bottom:** Relating to elevation, the closer a sampling point is to the bottom of a hill, the higher
       the Cs-137 levels.

4)  **Surface plan curvature, topographic wetness index (TWI):** Cs-137 migrates through the subsurface in soils by
infiltration. Thus, where water pools, at the flat or concave surface, Cs-137 levels are higher and have migrated to a
       further depth (Schimmack et al., 1994; Schimmack et al., 1989; Teramage et al., 2014). In general, slope degrees and
       curvatures – particularly profile and plan curvatures – are considered some of the most influential factors in hydrology
       and soil transport on a sloped surface; hence, they also influence Cs-137 movement (Gessler et al., 1995; Heimsath
       et al., 1997; Momm et al., 2012; Moore et al., 1993; Tesfa et al., 2009). Topographic wetness is one of the two most
powerful and most frequently used surface hydrology indices, along with the stream power index (Hengl and Reuter,
       2008).

5)  **Vegetation cover, hillslope aspect (categorical parameters):** If topographic factors and soil properties do not
       explain Cs-137 concentrations, other factors involved could be involved, such as vegetation cover (Coughtrey et al.,
       1989; Hashimoto et al., 2013; Korobova et al., 2007; Wakai et al., 2019) and locational differences (e.g., opposite
sides of the main channel with opposite aspects) (Gellis and Walling, 2011; Istanbulluoglu et al., 2008; Wicherek and
       Bernard, 1995).

In the current study, Cs-137 values were reported in two units: Bq kg$^{-1}$ (absolute measurement) and Bq m$^{-2}$ or mass depth (Bq
kg$^{-1}$ × dry bulk density × sample thickness). Since soil bulk density varies across samples, mass depth indicates the functionality
of soil bulk density as an explanatory variable, and it also provides a means to compare concentration levels among samples
with varied soil bulk densities (Kato et al., 2012; Miyahara, 1991; Rosén et al., 1999).

A preliminary check of autocorrelation among topographic parameters revealed that elevation and distance from the hillslope
bottom were highly correlated because of the vertical spatial configurations of sample locations. Thus, distance from the
hillslope bottom was removed from parameter selection and, in total, five topographic parameters, two soil properties, and two
categorical parameters were used in the subsequent analysis (Table 5).

**Table 5. List of data and parameters used in this study.**

| Data and parameters categories | Description |
|---|---|
| Cs-137 | Bq kg$^{-1}$, Bq m$^{-2}$ |
| Topographic parameters | elevation (m), slope degrees, upslope distance (m), plan curvature, TWI |
| Soil properties | water content (%), bulk density (g cm$^{-3}$) |
| Categorical | vegetation cover types, aspect |



### 4.3 Parameter Extractions

#### 4.3.1 Elevation

The elevation of sampling locations was extracted from the DEMs using the raster::extract function of R (R Core Team, 2015).

#### 4.3.2 Slope Degrees, Plan Curvature, Upslope Distance

SAGA GIS (Conrad et al., 2015) was used to compute slope degrees and plan curvature. SAGA was chosen because of its multiple options for the curvature calculation setting. The method used for plan curvature calculation was the second-order polynomial based on elevation values in nine surrounding cells (Zevenbergen and Thorne, 1987). The "D-Infinity Distance

Up" tool of TauDEM (Tarboton, 1997) was used to compute the upslope distance (Tarboton, 1997). This tool calculates the distance from each grid cell up to the ridge cells according to the reverse D-infinity flow directions. Multiple calculation options for upslope calculations are provided in the tool. In this study, both "minimum" distance and "surface" distance calculation options were used.

#### 4.3.3 TWI

TauDEM was used to calculate TWI values. The TWI tool calculates the ratio of the natural log of the specific catchment area (contributing area) to the slope. The index reflects water's tendency to accumulate at any point in the catchment and gravitational forces' tendency to move the water downslope (Quinn et al., 1991). No data values occur in locations where the slope is 0. TWI was calculated as follows (Quinn et al., 1991; Quinn et al., 1995) (Eq. 1):

$$TWI = \ln\left[\frac{A}{\tan(\beta)}\right] \tag{1},$$

where $TWI$ is the topographic wetness index, $A$ is the total upslope area accumulated in the current cell, and $\beta$ is the hillslope gradient. All topographic calculations in SAGA and TauDEM were saved as .TIF files. Then, the "raster::extract" function of R was again used to extract each topographic value for sampling locations.

#### 4.3.4 Vegetation cover type

For vegetation data, color winter imagery was downloaded from Google Earth (source image provided by Maxar

Technologies). The date of the imagery was 9 January 2017. Vegetation categories (evergreen or deciduous) were classified using the ArcGIS 10.7.1 image classification tool (ESRI Inc., 2019).

#### 4.3.5 Sample grouping by hillslope aspects

Based on the flow direction map (Fig. 5e), soil samples were divided into two groups: samples on the east side hillslopes (southwest facing) and samples on the west side hillslopes (northeast facing) of the main channel. Although the aspect at each



sample location (raster cell) might differ even on the same side, authors use a simple categorization due to the small number

of samples.

## 5 Methods

The selection of sampling locations in this study was influenced by the terrain or accessibility, and this selection was not

random or uniform. Thus, using distance-based spatial analysis – such as a semivariogram – was determined not to be

appropriate, and a numerical regression approach was adopted.

### 5.1 Statistical analysis for the effects of topographic parameters

#### 5.1.1 General additive model (GAM)

As a preliminary step, regression methods (linear, polynomial, logarithm, and bi-splines) were tested on Cs-137 measurements

in both Bq kg$^{-1}$ and Bq m$^{-2}$ units, using the five topographic parameters. The results showed that the polynomial and bi-splines

methods returned higher $R^2$ values than the other methods. Resulting plots showed that the fitting curves had multiple knots

that generated waveforms. Meanwhile, descriptive statistics indicated the exponential distribution and outliers of Cs-137 values

(see Sect. 6.1). All parameters had different ranges of numbers and precisions. Thus, a more flexible regression method was

sought, and the authors decided to use the general additive model (GAM) (Clark, 2013; Tesfa et al., 2009; Wood, 2017). GAM

is a generalized multiple linear regression involving a sum of smooth functions of covariates or fitting functions. Eq. 2 below

explains the basic concept of the GAM model (Wood, 2012).

$$g(\mu_i) = A_i\theta + f_1(x_{1i}) + f_2(x_{2i}) + f_3(x_{3i}) + \cdots f_k(x_{ki}) \tag{2},$$

where $x_1, x_2, \ldots, x_k$ are explanatory variables (predictors); $\mu_i \equiv \mathbb{E}(expected)(Y_i)$ and $Y_i \sim EF(\mu_i, \phi)$, $Y_i$ are response

variables; $EF(\mu_i, \phi)$ denotes an exponential family distribution with mean $\mu_i$ and a scale parameter, $\phi$; $A_i$ is a row of the

model matrix (parametric); and $\theta$ is a corresponding parameter vector. Finally, $f_j$ are smooth functions (non-parametric) of

the covariates of $x_k$.

For actual calculations and data output, the R packages mgcv and gamair (Wood, 2012) were used. GAM attempts to

simultaneously minimize overfitting and underfitting by calculating multiple coefficients in a model. Users can select a

smoothing function (basis) for the entire model or for each variable. They can also select fitting methods ("method") and the

number of knots ("k"), a distribution model type ("family"), and a relationship between means of variables ("link"). Adding a

categorical term ("identity") to the model is possible. GAM outputs parametric coefficients, $R^2$, p-values, a deviance-explained

percentage, and generalized cross-validations (GCVs) (Eq. A3) for the model, as well as the approximate significance of

smooth terms for each variable.





Although GAM has a weakness in that users cannot retrieve an individual regression expression from the output, it has been
used in soil and Cs-137 research (Linnik et al., 2020; Tesfa et al., 2009). This study used a spline-based smooth term for each
parameter (variable). The GAM setting was consistent for all model runs (see Appendix A-B).

Following the single-parameter GAM analysis, combinations of parameters were tested with GAM to determine the degree to
which these parameters explain Cs-137 concentrations. Combinations with more than three parameters were not tested to avoid
overfitting and to identify parameters that have a distinct influence on Cs-137 levels. The total number of combinations was
63 – the permutation of seven parameters with five topographic parameters and two soil properties, not considering the
parameter orders. Thus, the models included those with only topographic parameters, those with only soil properties, and those
in which both parameters were mixed.

To reflect the vertical profile of Cs-137 concentrations, GAM models were run against each depth layer. Thus, for topographic
parameters with one distinct value for the entire core sample, such as elevation and slope degrees, the runs employed a "one
to many" relationship (topographic parameters to Cs-137 in multiple depth layers). For soil properties with a measurement for
each depth layer, the runs employed a "one to one" relationship (soil property in a depth layer to Cs-137 in the depth layer).

### 5.1.2 GAM checks

The accuracies of these predictions were checked via the statistical indices that GAM returns with a "gam.check ()" function.
Outlier samples, which influenced predictions, were checked with Cook's distance (Cook, 1977) (Eq. 3). Stevens (1984) stated,
"Cook's distance measures the joint (combined) influence on the case being an outlier on y and in the space of the predictors."

$$D_i = \frac{\left(\hat{\beta}_{(-i)}-\hat{\beta}\right)'\text{X}'\text{X}\left(\hat{\beta}_{(-i)}-\hat{\beta}\right)}{ps^2} \quad i = 1, 2, \dots n \text{ (Cook, 1977)} \tag{3},$$

where $D_i$ is Cook's distance, $\hat{\beta}$ is the estimated regression coefficient, $'$X is the transpose matrix of n × p known constants, $\rho$ is
the number of predictors, and $s^2$ is the variance.

### 5.2 The effects of vegetation cover and hillslope aspects

To determine the effects of vegetation and locational differences on Cs-137 level predictions, an interaction term ("by=") was
added to the GAM models since vegetation and locational differences are categorical data (e.g., evergreen or deciduous areas).
Then, the significance of the models was compared pair-wise with a Tukey's honestly significant difference (HSD) post hoc
test (Tukey, 1949). This test runs an analysis of variance (ANOVA) and then compares the means of parameter pairs (Eq. 4).

$$q = \frac{m_1-m_2}{sm} \text{ (Tukey, 1949)} \tag{4},$$





where $q$ is the Tukey HSD result, $m_1$ is the larger of the means compared, $m_2$ is the smaller of the means compared, and sm

is the standard deviation of the means. R's "TukeyHSD ()" function returns the differences in means, confidence levels, and

adjusted p-values for each pair. The confidence interval in this study was 0.95. When the resulting interval overlaps the zero

point of the confidence interval, the difference between the pair is not significant.

### 5.3 Spatial extrapolation

Since practical decontamination planning requires a contamination estimate for an entire target area, it is important to

determine whether the best performance model works for the spatial estimation of Cs-137 levels. As the last step, a selected

GAM model was extrapolated to a basin in the study area, and the results were compared with the initial fallout deposition

estimated by the Japanese government and the United States Department of Defense (MEXT, 2011). For this step, GAM's

"predict()" function loaded topographic parameters in stacked raster files and predicted Cs-137 values in each raster cell based

on the selected model. The "predict ()" function was repeated for each depth layer. Then, the total Cs-137 concentration, down

to 30 cm deep, was calculated by summing the Cs-137 values in all layers.

## 6 Results

### 6.1 Cs-137 concentration levels

This section overviews the Cs-137 levels in the soil samples. Going forward, Cs-137 values in this article are decay-normalized

to 28 June 2016 (Eq. A1 and  Eq. A2).

Figure 8 shows the distributions of core total Cs-137 values in kBq kg$^{-1}$ (red) and kBq m$^{-2}$ (gray), as well as their means. The

y-axis is the number of core samples in each Cs-137 value bin (200 k). When Cs-137 was measured in kBq kg$^{-1}$, the Cs-137

values concentrated in six bins, and 45 % of the core samples were in the 200–400 kBq kg$^{-1}$ bins. In the mass depth unit (kBq

m$^{-2}$; gray), the distribution curve became flatter. In both cases, the above-average values had longer tails. The average total

core Cs-137 values among these samples were 261.3 kBq kg$^{-1}$ and 1,082 kBq m$^{-2}$. The same data on a log scale (Fig. 9) shows

that the distributions become close to a normal distribution; however, the curves are not smooth.





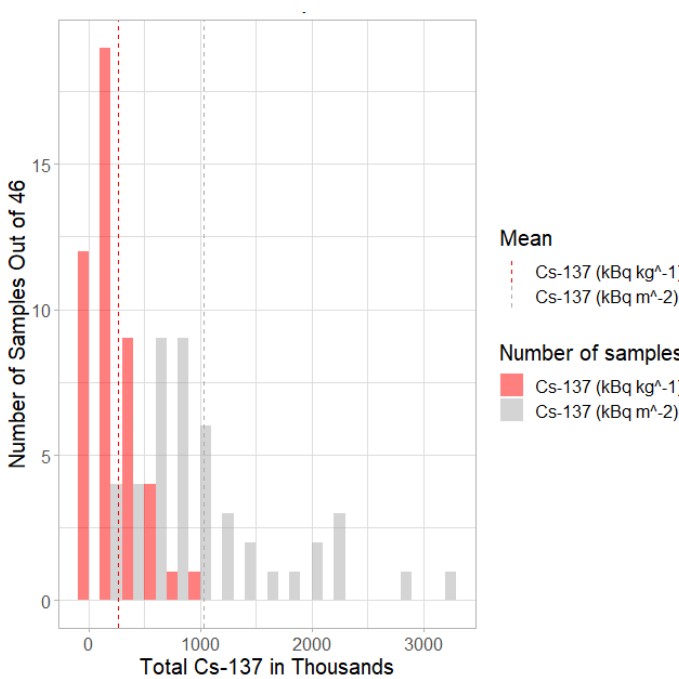

**Figure 8. Distribution of total Cs-137 in a core sample (red: Cs-137 kBq kg$^{-1}$; gray: Cs-137 kBq m$^{-2}$). A unit of each bin is 200 k.**

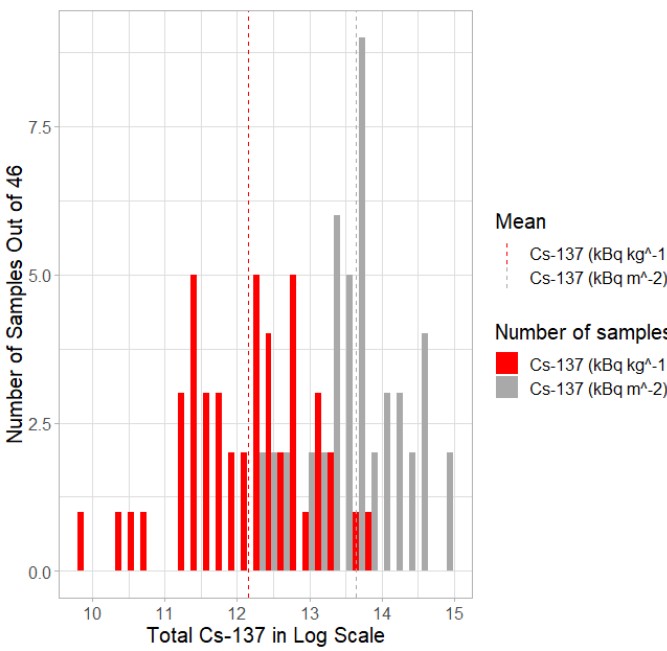

**Figure 9. Distribution of total Cs-137 in a core sample in a log scale (red: Cs-137 kBq kg$^{-1}$; gray: Cs-137 kBq m$^{-2}$).**

Depth distributions of Cs-137 in both kBq kg$^{-1}$ and kBq m$^{-2}$ (Fig. 10) show exponential vertical decreases, similar to the results

reported in previous Cs-137 research (Fujii et al., 2014; Koarashi et al., 2012; Matsunaga et al., 2013; Takahashi et al., 2015;





Tanaka et al., 2012; Teramage et al., 2014). These samples were collected five to seven years after the FDNPP accident, and they still held large portions of Cs-137 in the top layers. The average depth of 90 % concentration (90 % of the whole Cs-137 in a core sample) was 7.45 cm. Standard deviations in the upper layers were large, up to 108.1 kBq kg$^{-1}$ and 317.7 kBq m$^{-2}$ in the top 2 cm (red dots in Fig. 11 and Fig. 12). However, the COVs were below 100 % in the top layer (gray dots in Fig. 11 and

Fig. 12). The COVs increased toward the 14–16 cm depths (Fig. 11 and Fig. 12). The bottom layer (27.5-30.0 cm depth) shows the largest COVs because the deepest layer has a smaller number of samples. At some locations, the sampling tube hit tree roots or rocks, or soil dropped when the tube was pulled out of the ground.

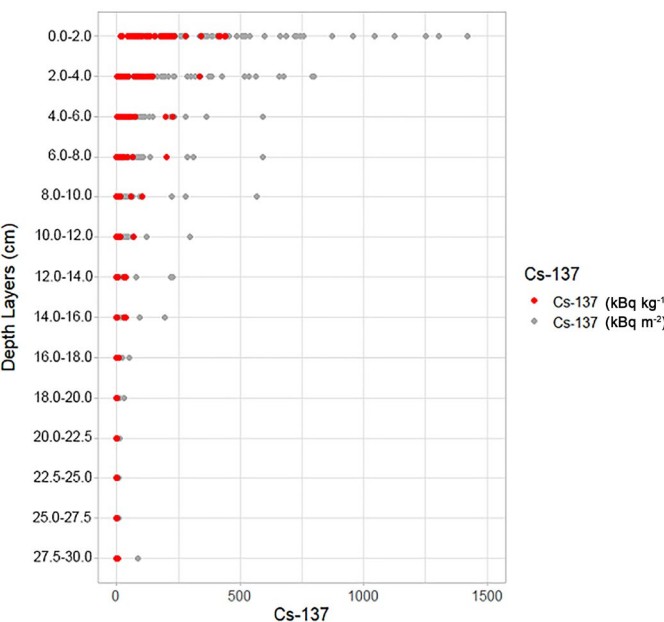

**Figure 10. Depth distribution of Cs-137 in kBq kg$^{-1}$ (red) and kBq m$^{-2}$ (gray).**





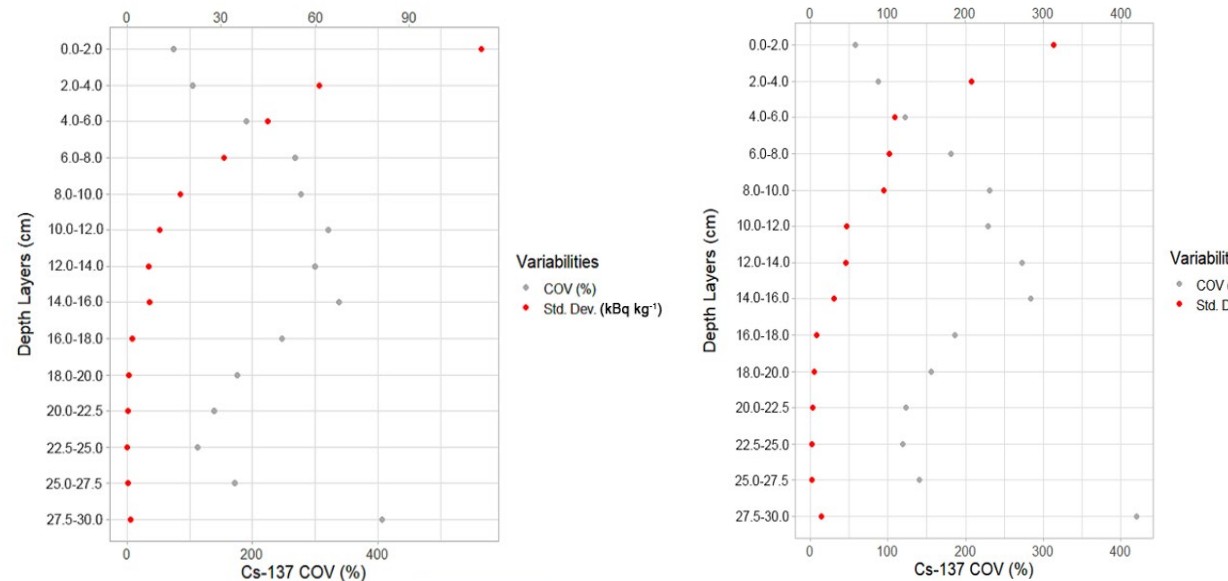

**Figure 11. Standard deviations (red) and COVs (gray) by depth in kBq kg⁻¹.**

**Figure 12. Standard deviations (red) and COVs (gray) by depths in kBq m⁻².**

## 6.2 Test 1 results: Cs-137 accumulation patterns on a simple representative hillslope

This section displays the Cs-137 accumulation levels on a simple representative hillslope. The following three graphs (Fig. 13–Fig. 15) organize the total core Cs-137 concentrations of all samples by one topographic parameter (elevation in meters). In Fig. 13–Fig. 15, red circles are added to indicate the four highest concentration samples and to compare their Cs-137 values and elevations (Fig. 13), slope degrees (Fig. 14), and migration head depths (Fig. 15).





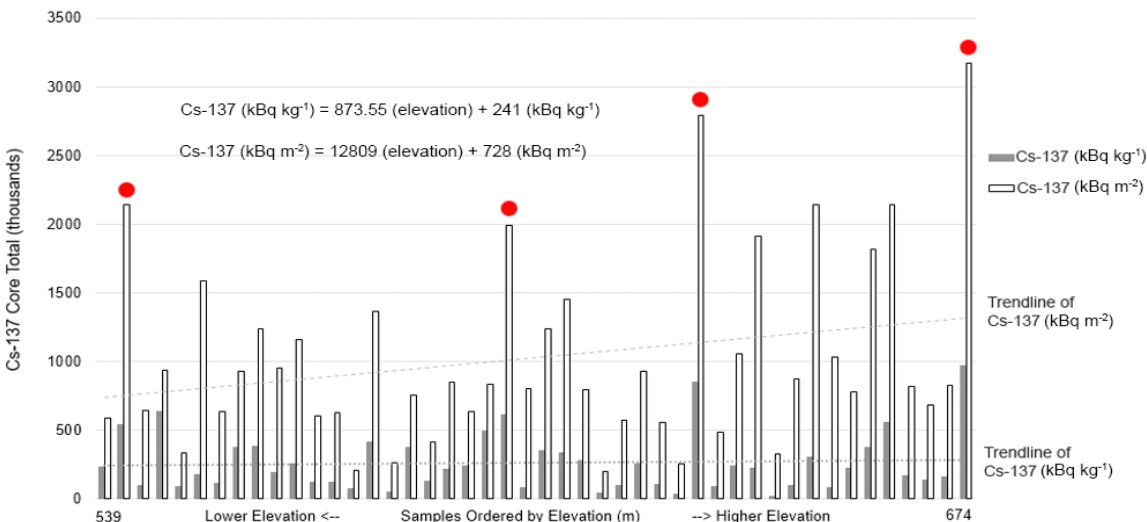

**Figure 13. Core total Cs-137 concentrations in both units along elevation (m).**

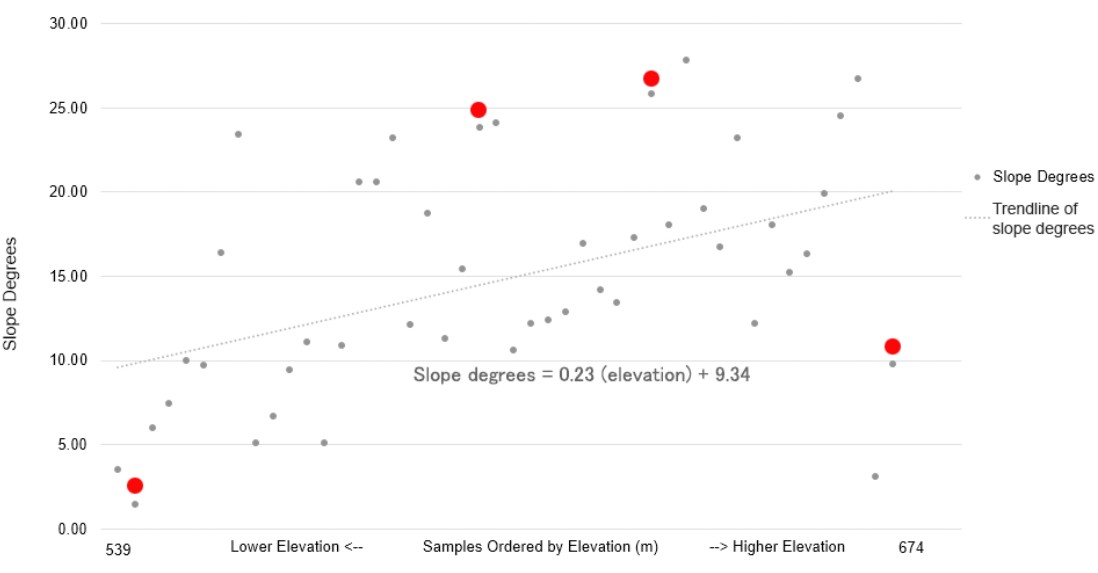


**Figure 14. Slope degrees of sampling points along elevation (m).**



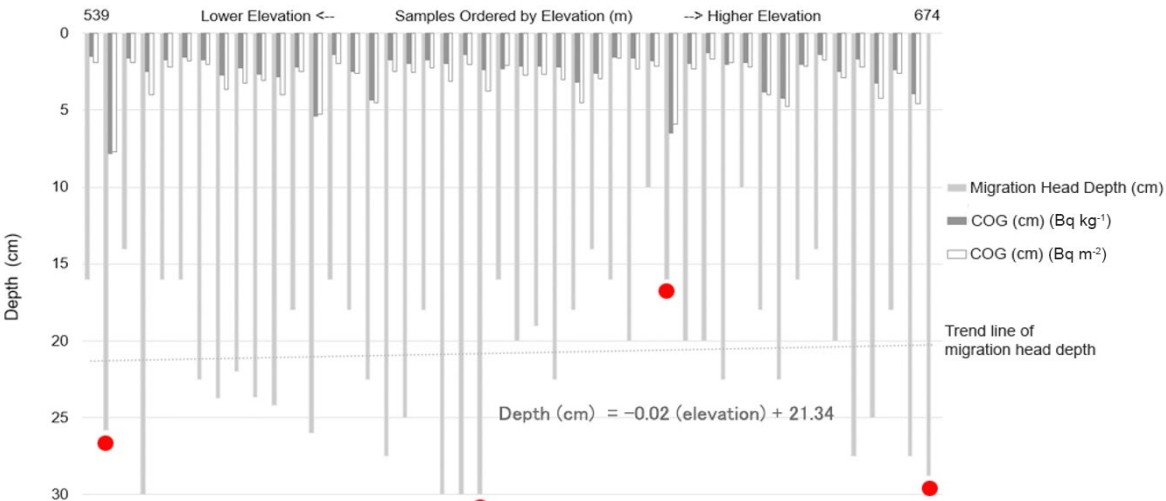

**Figure 15. Migration head depths (cm; the depth at which Cs-137 decreased to 100 Bq kg⁻¹ in each core sample) and the depths of COGs (cm) in each core sample for both units (Bq kg⁻¹ and Bq m⁻²).**


In Fig. 13, both trend lines of Cs-137 values show upward trends along elevation. However, the actual Cs-137 level presents a "camelback" shape. Figure 14 shows slope degrees at each sampling location along elevation. The trend line indicates an increase in slope degrees along elevation. However, the Pearson's correlation index between elevation and slope degrees was 0.24 due to the outliers at the ridge top and at the hill bottom.

Figure 15 graphs the migration head depth (cm) and the center of gravity (COG) depth (cm) for each core sample. The conservative background contamination level of the study site was 100 Bq kg⁻¹. Thus, the depth at which Cs-137 decreased to 100 Bq kg⁻¹ was considered the deepest level to which the FDNPP accident–derived Cs-137 had migrated downward. In four samples, the migration head had already reached 30 cm deep (the length of the sampling tube) in 2016. The average depth of the migration head was 20.79 cm.

In Fig. 15, the COG depth was added to show relative downward migration rates. COG depth is not the depth of the largest Cs-137 concentration in the core sample; rather, it is the depth of the center of the Cs-137 concentration density at 30 cm deep. It is calculated as below (Shiozawa et al., 2011) (Eq. 4).

$$(x) = \frac{\sum_i x_i C_i \Delta x_i}{\sum_i C_i \Delta x_i} \tag{4},$$

where $x_i$ is the middle depth of each depth layer, $C_i$ is the Cs-137 concentration of each depth layer, and $\Delta x_i$ is the depth of
each depth layer. The average COG depths in Bq kg⁻¹ and Bq m⁻² were 2.53 cm and 3.02 cm, respectively. Both migration head depths and COV depths were deeper at lower elevations. However, as with the core total of Cs-137 in Fig. 13, the actual





measurement showed a camelback shape along elevation. According to a Pearson's correlation index, core total Cs-137 concentration levels in Bq kg$^{-1}$ and the migration head depths were correlated (0.68), and Cs-137 in Bq m$^{-2}$ and migration head depths were weekly correlated (0.45). The COG depths did not show significant correlations with the core total of Cs-137, indicating a discrepancy between the deepest reach of contamination and the relative depth of the highest concentration after several years since the initial fallout.

### 6.3 Test 2 Results: The effects of topographic parameters on Cs-137 accumulation patterns

### 6.3.1 Parameter ranges by DEM resolutions

This section displays the differences between topographic parameter values extracted from the 1 m and 10 m DEMs. Figure 16 shows the value ranges of each topographic parameter by DEM resolution. The numbers on the top row under each graph indicate COVs. The numbers on the bottom rows are the differences of medians between two DEMs divided by the range of all values of the parameter. As evidenced in the graphs, DEM resolution differences caused considerable data-range variations in the upslope distance and TWI. Plan curvature displayed a smaller disparity between the DEMs, but the range of values was approximately four times larger with the 1 m DEM compared to the 10 m DEM, while the COV was 6.5 times larger with the 10 m DEM.





**Figure 16. Topographic parameter distributions. Two box plots are presented for parameters affected by different DEM resolutions.**

### 6.3.2 GAM results: Single parameter

This section reports the GAM results of the single-parameter models to determine each parameter's explanatory power for Cs-137 values. In all instances of GAM, each model was run against each depth layer, and the averages of deviance explained percentages throughout the depths were calculated (Table 6). For p-values, since there is a clear cut point for significance ($p \leq 0.05$), the number of depth layers, whose p-values were equal or less than 0.05, are reported (e.g., "4/14" means that four depth layers' p-values were $\leq 0.05$.). Table 6 includes the results of topographic parameters, which were affected by the DEM

resolutions, while Table 7 displays the results of soil property parameters, which were unaffected by the DEM resolutions. None of the single parameters in either table returned deviance-explained percentages above 35 % (which was found via a model including water content). The lowest p-value was 0.05, which was significant and also found via a model with water





content. Among the topographic parameters, TWI was the most effective with the 1 m DEM, while slope degrees were the
most effective with the 10 m DEM. Soil properties, especially water content, demonstrated higher explanatory power than
topographic parameters. However, although the p-values of soil properties were significant in more than ten depth layers in
Bq kg$^{-1}$ units, their significance decreased in Bq m$^{-2}$ units. All parameters showed slightly higher explanatory power in Bq kg$^{-1}$ than in Bq m$^{-2}$, except for plan curvature extracted from the 10 m DEM.

**Table 6. The results of single topographic parameter GAM models for deviance-explained percentages and p-values (the numbers**
**are the averages of all depth layers). "( )" indicates standard deviation.**

| 1 m DEM | | | | |
|---|---|---|---|---|
| Parameter | Deviance explained (%) | # of layers: p-value ≤0.05 | Deviance explained (%) | # of layers: p-value ≤0.05 |
| | Bq kg$^{-1}$ | | Bq m$^{-2}$ | |
| elevation (m) | 20.96 (14.10) | 1/14 | 20.18 (12.48) | 0/14 |
| upslope dist. (m) | 12.90 (14.02) | 0/14 | 8.52 (19.99) | 0/14 |
| slope degrees | 19.41 (23.00) | 2/14 | 9.09 (18.63) | 1/14 |
| plan curvature | 16.81 (14.19) | 2/14 | 15.73 (11.75) | 0/14 |
| TWI | 26.61 (22.21) | 2/14 | 21.02 (18.68) | 1/14 |
| 10 m DEM | | | | |
| Parameter | Deviance explained (%) | # of layers: p-value ≤0.05 | Deviance explained (%) | # of layers: p-value ≤0.05 |
| | Bq kg$^{-1}$ | Bq m$^{-2}$ | Bq kg$^{-1}$ | Bq m$^{-2}$ |
| elevation (m) | 23.19 (12.12) | 1/14 | 20.17 (10.98) | 2/14 |
| upslope dist. (m) | 22.76 (17.31) | 3/14 | 18.10 (17.92) | 1/14 |
| slope degrees | 29.46 (18.73) | 7/14 | 23.25 (17.84) | 5/14 |
| plan curvature | 3.64 (5.00) | 0/14 | 4.78 (7.71) | 0/14 |
| TWI | 25.63 (19.45) | 1/14 | 22.41 (20.91) | 1/14 |



**Table 7. The results of single-soil-property-parameter GAM models for deviance-explained percentages and p-values (the numbers are the averages of all depth layers). "( )" indicates standard deviation.**

| Parameter | Deviance explained (%) | # of layers: p-value ≤0.05 | Deviance explained (%) | # of layers: p-value ≤0.05 |
|---|---|---|---|---|
| | Bq kg$^{-1}$ | | Bq m$^{-2}$ | |
| water content (%) | 35.21 (21.01) | 11/14 | 28.01 (24.13) | 9/14 |
| bulk density (g cm$^{-3}$) | 33.31 (15.66) | 10/14 | 18.26 (16.68) | 5/14 |


### 6.3.3 GAM combination parameters results

Figures 17–22 show the percentages of deviance explained by depth layers, including one, two, and three-parameter models. In these figures, deviance-explained percentages were used instead of p-values because the authors wanted to determine the extent to which the variability of Cs-137 values was explained, rather than to test a simple model significance. The deviance-

explained percentages and p-values were in accordance; that is, higher explanation percentages returned lower p-values. In all six figures, the red dots represent the explanation percentages of Cs-137 in Bq kg$^{-1}$ units and the gray dots represent the percentages of Cs-137 in Bq m$^{-2}$ units.

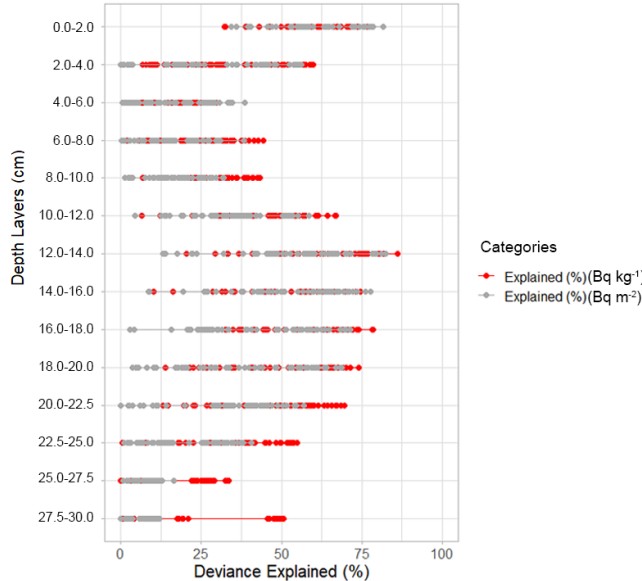

**Figure 17. Deviance explained (%) using the 1 m DEM.**





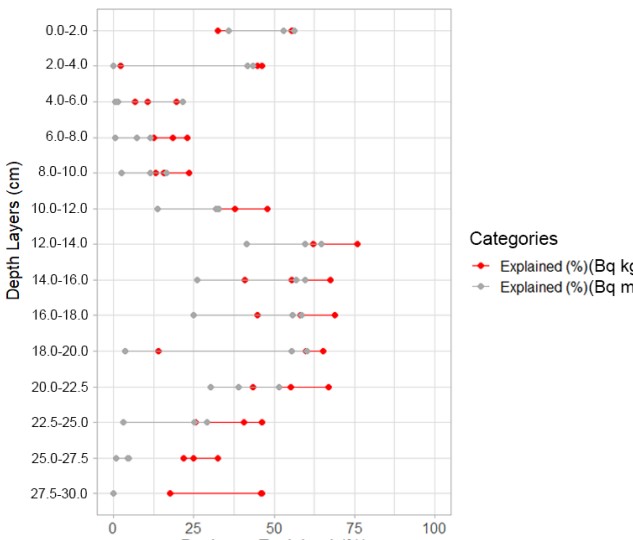

Figure 18. Deviance explained (%) using the 1 m DEM with soil properties only.

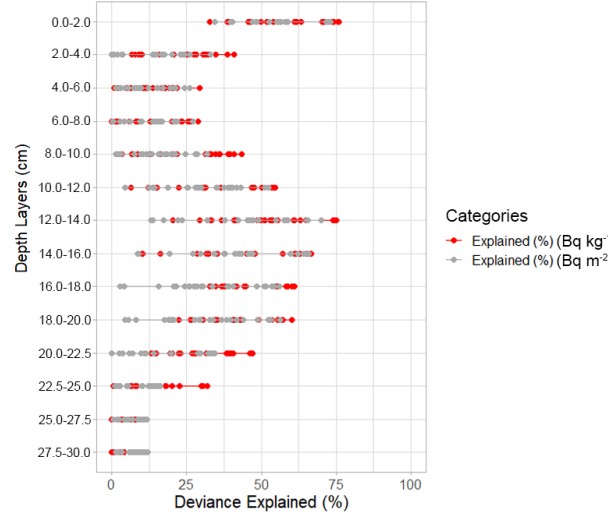

Figure 19. Deviance explained (%) using the 1 m DEM with topographic parameters only.

Figure 17 includes the results of using the 1 m DEM for all 63 combinations. The next two figures present the deviance-explained percentages using the 1 m DEM with only water content and bulk density (Fig. 18) and with only topographic parameters (Fig. 19). Figure 20 includes the results of using the 10 m DEM with all 63 combinations. The next two figures show the percentages using the 10 m DEM with only water content and bulk density (Fig. 21) and with only topographic parameters (Fig. 22).

Overall, the models using Bq kg$^{-1}$ (red dots) returned better predictions than those using Bq m$^{-2}$ (gray dots), and the models with the topographic parameters extracted from the 10 m DEM performed better than those using the 1 m DEM. However, in the topographic models based on the 10 m DEM, the low-end outliers became distinct (see gray dots in Fig. 17–Fig. 22).

All six figures showed similar vertical S-curves. The deviance-explained percentages of some models were over 75 % for the top layer (0.0–2.0 cm). The percentages declined below the top layer and rose again around a depth of 10.0–20.00 cm. This S-shape resembles the vertical profiles of the COVs for water content and bulk density (Table 4) and the COVs of the Cs-137 measurements (Fig. 11–Fig. 12). Models with only soil properties and models with only topographic parameters returned similar vertical curves (Fig. 18–Fig. 19; Fig. 21–Fig. 22).





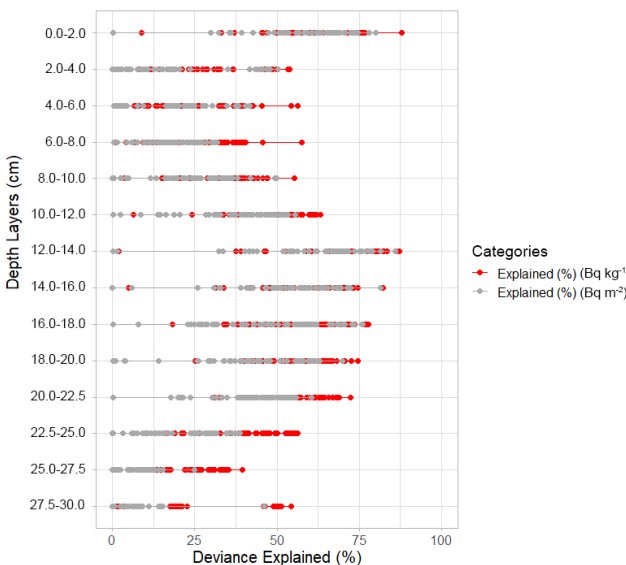


**Figure 20. Deviance explained (%) using the 10 m DEM.**

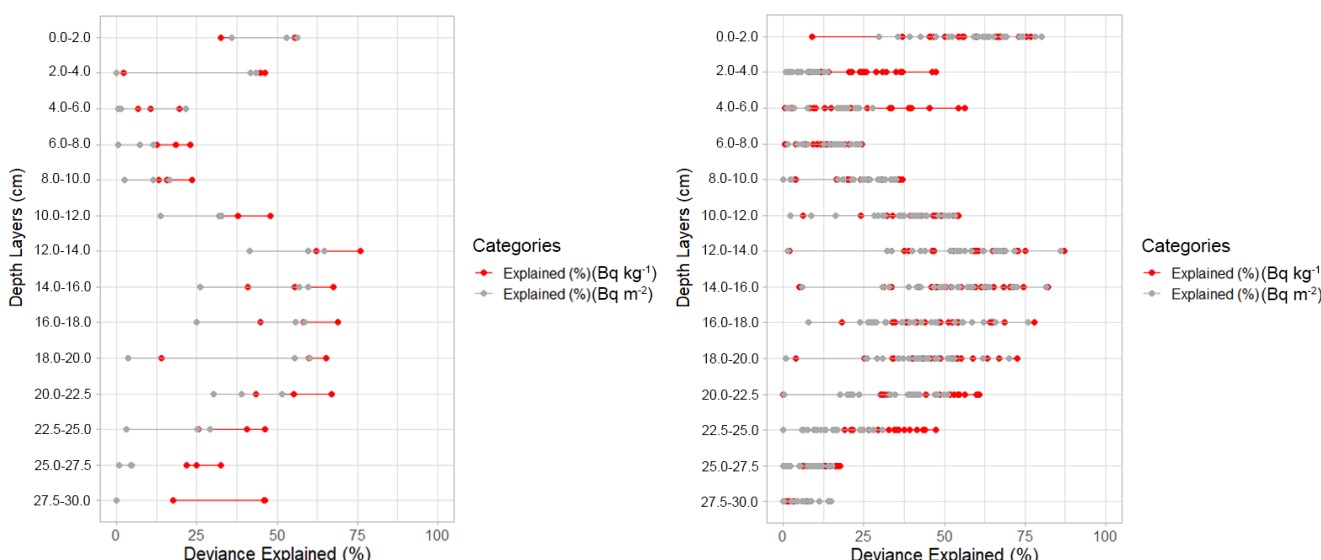

**Figure 21. Deviance explained (%) using the 10 m DEM with soil properties only (same graphics as Figure 18. Repeated for comparison purpose).**

**Figure 22. Deviance explained (%) using the 10 m DEM with topographic parameters only.**





**Table 8. The most effective parameter combinations with the best deviance-explained percentages and p-values (the numbers are the averages of all depth layers). "( )" indicates standard deviation.**

| 1 m DEM | | | | | | | |
|---|---|---|---|---|---|---|---|
| | Permeameter combinations | Core average deviance explained (%) | # of layers: p-value ≤0.05 | | Parameter combinations | Core average deviance explained (%) | # of layers: p-value ≤0.05 |
| Bq kg⁻¹ | | | | Bq m⁻² | | | |
| All | water content + bulk density + elevation | 54.92 (std. dev. 18.72) | 6/14 | All | water content + bulk density + elevation | 44.32 (std. dev. 23.51) | 5/14 |
| Topo* only | elevation + plan curvature + TWI | 40.11 (std. dev. 22.34) | 0/14 | Topo only | elevation + plan curvature + TWI | 36.80 (std. dev. 23.82) | 0/14 |
| 10 m DEM | | | | | | | |
| Permeameter combinations | Core average deviance explained (%) | # of layers: p-value ≤0.05 | | Permeameter combinations | Core average deviance explained (%) | # of layers: p-value ≤0.05 | |
| Bq kg⁻¹ | | | | Bq m⁻² | | | |
| All | water content + bulk density + elevation | 55.98 (std. dev. 17.63) | 6/14 | All | water content + bulk density + elevation | 45.09 (std. dev. 22.74) | 5/14 |
| Topo only | elevation + slope degrees + upslope distance | 48.62 (std. dev. 26.18) | 4/14 | Topo only | elevation + slope degrees + upslope distance | 43.93 (std. dev. 27.37) | 5/14 |

*An abbreviation of "Topography".

Table 8 lists the most effective parameter combinations for explaining Cs-137 values. The rows are separated into combinations with all the parameters (including soil parameters) and combinations with only the topographic parameters. For both 1 m and 10 m DEMs, the most effective combinations were "water content + bulk density + elevation." When soil properties were removed, the most effective topographic parameter combinations were "elevation + plan curvature + TWI" when using the 1 m DEM and "elevation + slope degrees + upslope distance" when using the 10 m DEM. Overall, model performance was higher using the 10 m DEM and Cs-137 in Bq kg⁻¹ units. The Cs-137 units made no difference in the best performance parameter combinations. The results showed improvement by combining topographic features. However, no combinations with topographic parameters reached statistically significant levels (p≤0.05).





### 6.3.4 GAM Prediction Results Validations

No reference data were available to validate the model results. Therefore, Cs-137 levels were reverse-predicted using the two best performance models with the 10 m DEM from Table 8, and the model fits were checked. The "predict.gam()" function was used to make the prediction, and then the performance of the models was checked with the "gam.check ()" command.

This command outputs several performance metrics of prediction fit. Figure 23 displays the check results for the two prediction models. The output metrics are deviance-explained percentages, $R^2$, Akaike information criteria (AIC) for maximum likelihood (Eq. A4), and generalized cross-validation (GCV), which is the average ability of models fitted to the remaining data to predict the omitted data.

Higher deviance-explanation percentages and $R^2$ values indicate that a model performs well. Lower AIC indicates a good

model with less information loss (parsimony), and lower GCV suggests a good model with a smaller prediction error (Hastie et al., 2009; Wood, 2017). The graphs in Fig. 23 show all four matrices' results by depth layer averages. The deviance-explained percentages and $R^2$ showed camelback-shaped fit changes across the depth. AIC and GCV decreased continuously.

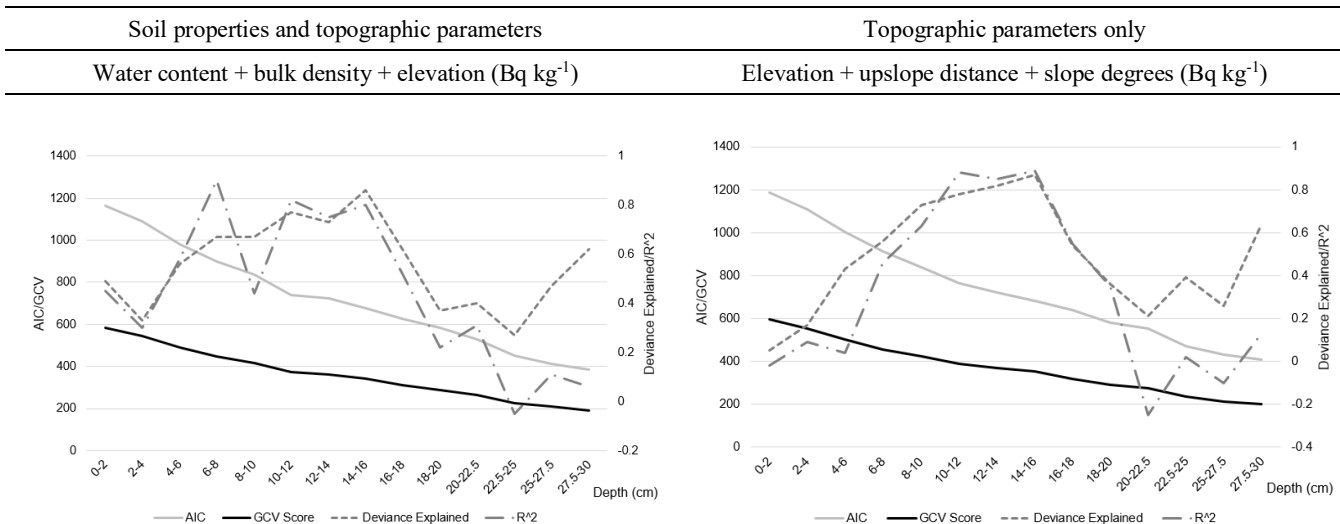

**Figure 23. Model fit diagnosis results of the best-performing models using the 10 m DEM in Bq kg⁻¹ unit (light gray: AIC; black:**
**GCV score; dashed: deviance-explained percentage: loosely dash-dotted: R²).**



### 6.3.5 GAM prediction results validations: Outlier/influential sample analysis

Figure 24 presents linear regression scatterplots of actual Cs-137 values versus predicted Cs-137 values for the same two
models. The model with soil properties returned a better model fit, with an $R^2$ of 0.62, compared to the model with only
topographic parameters, which had an $R^2$ of 0.25. In the bottom graphs of Fig. 24, the *y*-axis is the Cook's distance and the *x*-
axis is leverage (i.e., the measure of the extremity of a predictor). The red contours in the Cook's distance plots are standardized
residual contours. The white circles, which are away from other sample groups, have a greater influence on the entire data
distributions.


| With soil properties | Topographic parameters only |
|---|---|
| Water content + bulk density + elevation (Bq kg$^{-1}$) | Elevation + upslope distance + slope degrees (Bq kg$^{-1}$) |

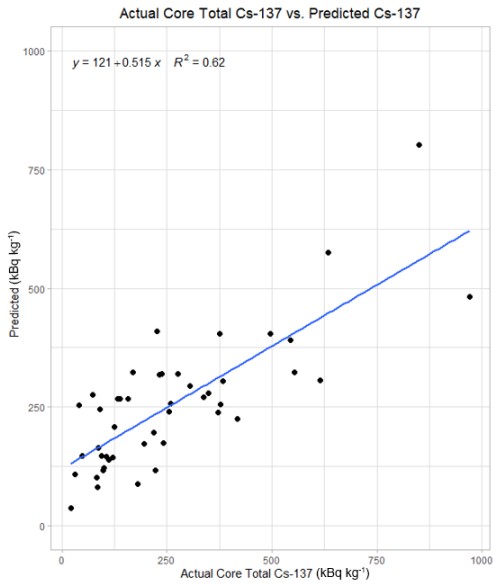 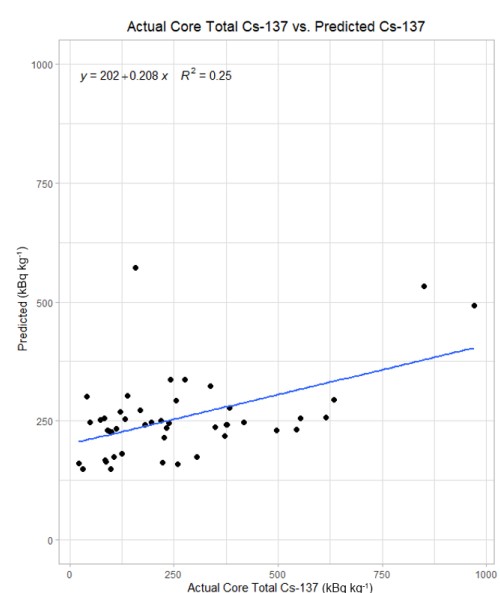





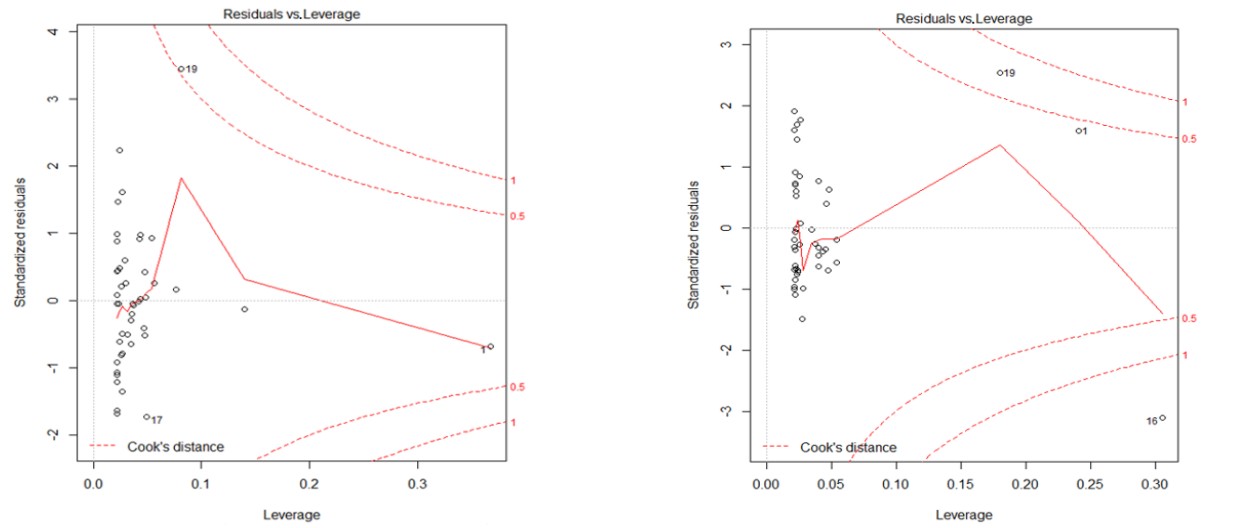

**Figure 24. Regression analysis of actual and predicted Cs-137 values and outliers (top row graphics: regression analysis; bottom row graphics: Cook's outliers).**

These influential outlier samples are plotted in Fig. 25 with red circles over the flow direction map (left) and the winter

vegetation map (right) of the study site. The locations of these outliners were spread on opposite sides of the hillslopes (Fig. 25, left), and they were found in areas without evergreen vegetation (Fig. 25, right).

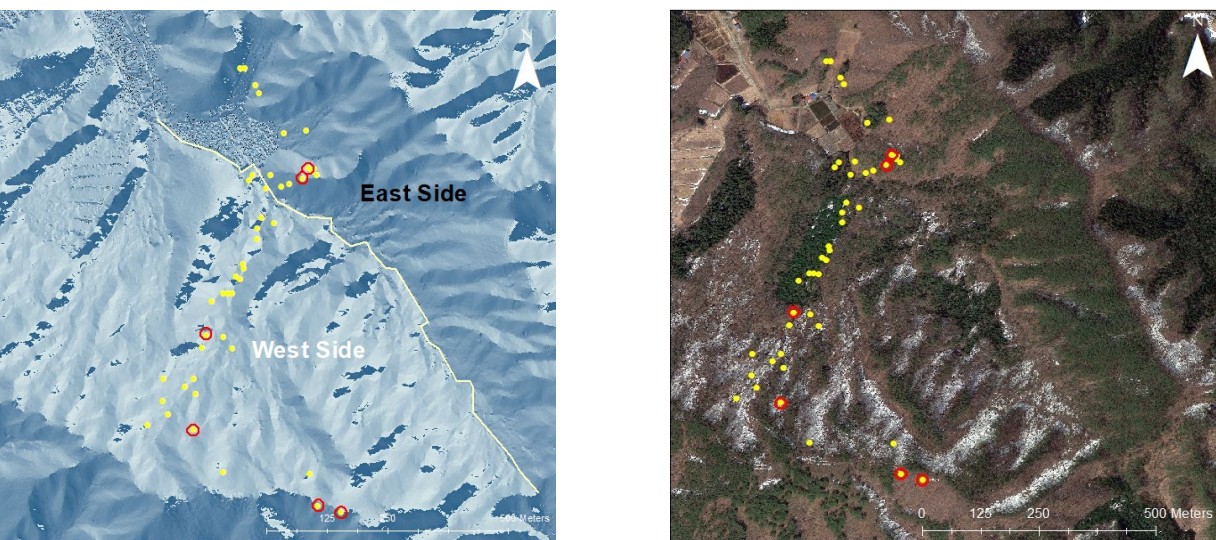

**Figure 25. Cook's distance outlier samples overlaid on a flow direction map (left) and a winter vegetation image (right) (map data: © Google, Maxar Technologies). On the flow direction map, the cream line indicates the line where the surface flow converges into**

**the 510 main channel from the hillslopes on the opposite side.**



### 6.4 Test 3 Results: Vegetation cover and hillslope aspects

To determine the significance of vegetation cover and locational differences on the Cs-137 prediction results, an interaction term ("by=") was added to the GAM prediction models, and the models were run again. Both hillslope aspect and vegetation
interaction terms had two categories: the east and west sides of a channel, and evergreen and deciduous vegetation cover differences. Then, the mean differences in Cs-137 prediction results were checked pair-wise using a Tukey's HSD test. Each pair contained a combination of one parameter plus one categorical term on both sides (e.g., water content on the east side vs. elevation on the west side, elevation in the evergreen area vs. upslope distance in the deciduous area). The same paring was tested for all depth layers, and a confidence interval range was displayed for that pair.


Table 9 shows five pairs that returned significant mean differences. The results indicated that the pairs with topographic parameters returned considerable mean differences in Cs-137 predictions, particularly when the prediction results were subgrouped by vegetation cover types. Among the topographic parameters, the upslope distance was the parameter that appeared to most affect the prediction results at the particular study site. Notably, the directions of confidential interval ranges between
the "upslope distance vs. elevation" pair (Table 9a) and "slope degrees vs. upslope distance" pair (Table 9b) were opposite. According to Fig. 14, slope degrees increase along with an elevation increase; however, they influenced Cs-137 predictions in opposite ways.

**Table 9. The parameter pairs that showed mean differences in the Tukey's HSD test for Cs-137 predictions (a 95 % confidence**
**interval did not overlap the zero point). The last column indicates the directions of differences from the 0.0 point of the confidential interval.**

| The parameter pairs returned significant mean differences | | | | | | Directions from 0.0 point of the confidential interval |
|---|---|---|---|---|---|---|
| water content | + | deciduous | vs. | elevation | + | evergreen | negative |
| upslope distance | + | evergreen | vs. | elevation | + | deciduous | positive [a] |
| upslope distance | + | evergreen | vs. | elevation | + | evergreen | positive [a] |
| slope degrees | + | deciduous | vs. | upslope distance | + | evergreen | negative [b] |
| slope degrees | + | evergreen | vs. | upslope distance | + | evergreen | negative [b] |

### 6.5 Test 4 Results: Spatial extrapolation test

In this section, using the GAM combinations with the best deviance-explained percentage, Cs-137 concentration in a large
basin (Fig. 5d) was extrapolated. According to Table 8, the most effective topographic parameter combination was "elevation + slope degrees + upslope distance" using the 10 m DEM in Bq kg$^{-1}$ units. First, the extrapolation was tested with the original GAM prediction models for each depth layer. However, the spatially extrapolated Cs-137 values were overblown in some depth layers, returning unrealistically large concentration values. To resolve this issue, k=3 (wiggliness) was manually set to constrain the model. The number of k= was increased incrementally; the resulting model performance and output plots were





checked, and the number three was selected. By adding the k= term, the deviance explained percentage of the GAM model decreased from the core average 48.62 (original) to 39.25; the standard deviation decreased from 26.18 to 17.19; and the p-value (0.19) did not change.

The extrapolation result (Fig. 26) shows that the Cs-137 concentration in the basin was 3,311 MBq kg$^{-1}$ (780 kBq m$^{-2}$) down to 30 cm deep as of 28 June 2016, assuming the dry bulk density to be 0.44 g cm$^{-3}$. Since this was an area-wide extrapolation, the calculation was done by ([total Cs-137 (Bq kg$^{-1}$)/ area (m$^{-2}$)] × bulk density [kg m$^{-3}$] × depth [m]). When the concentration was reverse-decay-corrected, the prediction became 3,741 MBq kg$^{-1}$ (880 kBq m$^{-2}$) for 15 March 2011. The estimated concentration was 12 % lower than the lowest end of the Japanese government's initial deposition estimate for the region (1,000–3,000 kBq m$^{-2}$; MEXT, 2011). The total includes the pre-FDNPP accident background contamination and the estimated depositions in all raster cells, which include channels. The widths of the channels in the forest are about 30 cm to 1 m. Thus, they were not separated in the 10 m resolution raster cells. The upper left (former rice paddy areas) and bottom right (lowland areas) corners of Fig. 26 show white stripes because upslope distance and TWI did not return values in the flatlands. Figure 26 also shows the overlaid positions of Cook's outliers (red circles).

The extrapolated values in Bq kg$^{-1}$ were overestimated toward higher values (+06 in the map legend; Fig. 26). One of the samples in this study at the highest ridge did contain a core total Cs-137 value of 1.36 MBq kg$^{-1}$. This sample appeared to have contributed to overestimation in the ridge areas. On the other hand, the calculated concentration in Bq m$^{-2}$ was lower than the government's initial estimate. The average bulk density of the collected samples was 0.44 g cm$^{-3}$, and the actual average density at the study site soils might have been higher since most of the samples were collected at accessible locations – not at locations with steeper slope degrees (i.e., locations difficult to access with tools). In the areas with steep slope degrees, soil density could have been higher. If the bulk density was 1.0g/cm$^3$, the Cs-137 concentration of the basin based on Bq kg$^{-1}$ would have been 1,770 kBq m$^{-2}$ in 2016.





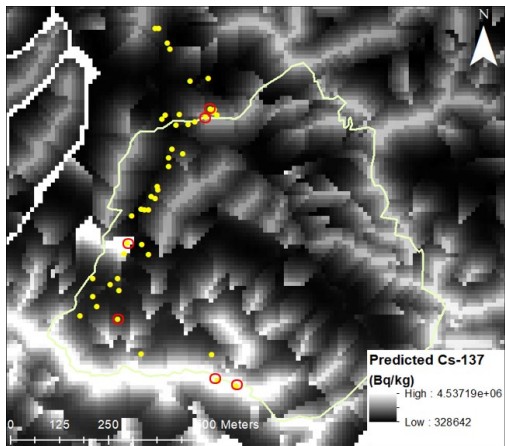

**Figure 26. Extrapolated Cs-137 (Bq kg⁻¹) using "elevation + slope degrees + upslope distance." (White indicates higher concentration.) Red circles are the location of Cook's outliers.**

## 7 Discussion

### 7.1 The effects of topography

Topographic effects on Cs-137 accumulation patterns are, by themselves, weak when a single parameter is used for Cs-137 prediction (deviance-explained percentages 3.64–29.46; Table 6). Elevation and slope degrees are commonly used in geomorphic assessment (Catani et al., 2010; Chen et al., 1997; Griffiths et al., 2009; Hoover and Hursh, 1943; Roering et al., 2001; Roering, 2008; Roering et al., 1999; Rossi et al., 2014; Yang et al., 1998) and in Cs-137 contamination research (Ritchie and McHenry, 1990; Walling et al., 1995). However, prediction models with only elevation or slope degrees explained Cs-137 concentrations at less than 29.46 % in either unit (Bq kg⁻¹ or Bq m⁻²) or either DEM (1 m or 10 m) employed in this research.

The explanation power of water content and dry bulk density was higher (deviance explained percentages 28.01–35.21) (Table 7). When topographic parameters were combined among themselves or combined with soil property data, the GAM models' explanation powers improved (the best explanation percentages ranged from 29.46 % for a single-parameter model to 55.98 % for a mixed-parameter model; Table 8).

Despite the fact that elevation showed no significance for explaining Cs-137 as a single parameter, it was the most prevalent topographic parameter (Table 8) among the models with higher explanation percentages. The average minimum upslope distance from sampling points at the study site was 16.90 m with the 1 m DEM and 68.52 m with the 10 m DEM (Fig. 16), indicating that those hillslopes are composed of multiple small sub-basins with less than 100 m minimum upslopes. Hence, in this particular case, elevation and slope alone could not efficiently explain all the Cs-137 concentration patterns. The camelback-shaped Cs-137 profiles on the representative hillslope (Fig. 13–Fig. 15) are the results of this spatial configuration.





## 7.2 Subsurface Migration

The vertical profiles of the GAM explanation percentages of soil properties and topographic parameters were similar throughout the depths (Fig. 17–Fig. 22), indicating that Cs-137 vertical profiles were the result of their co-functionalities. However, predictions solely with topographic parameters displayed wider ranges in those percentages and, for some samples, the models were unable to explain Cs-137 concentrations (gray dots closer to zero in Fig. 19 and Fig. 22).

Regarding Cs-137 accumulation on a sloped surface (Figures 13–15), it was generally assumed that: 1) Cs-137 accumulates at the bottom of the slope where surface runoffs converge, and 2) its accumulation levels are higher in lower slope-degree areas where surface water pools. According to the trendline in Figure 15, the migration head depths were deeper at lower elevations following the abovementioned assumption 1). Considering that the migration head depths had already reached 30 cm deep in four of the samples, the actual trendlines in Figure 15 could have been much steeper. However, contrary to the assumptions, Cs-137 accumulation levels were lower at lower elevations (Figure 13), and the locations with higher Cs-137 accumulations or deeper migration head depths were not necessarily the locations with the lowest slope degrees (Fig. 14).

**Table 6. Comparison of Cs-137 downward migration depth between the CNPP accident–affected area and Fukushima at about the same length of time after the nuclear plant accidents.**

| Ukraine, Belarus, Russia (Ivanov et al., 1997; 15 samples) | | | Current Study (Iitate Village, Fukushima, Japan; 58 samples) | | | |
|---|---|---|---|---|---|---|
| Depth of 90 % threshold of Cs-137 accumulation (Bq cm$^{-3}$) | Number and % of samples | | Depth of 90 % threshold of Cs-137 accumulation (Bq m$^{-2}$) | Number and % of samples | | |
| | Years passed | Years passed | | Years passed | Years passed | Years passed |
| | 7 | 8 | | 6 | 7 | 8 |
| 2 cm | 1 (6.7 %) | 1 (6.7 %) | 2 cm | | | |
| 4 cm | 5 (33.3 %) | 1 (6.7 %) | 4 cm | 8 (13.8 %) | 1 (1.7 %) | 11 (19.0 %) |
| 6 cm | 1 (6.7 %) | 1 (6.7 %) | 6 cm | 4 (6.9 %) | 1 (1.7 %) | 4 (6.9 %) |
| 7 cm | 4 (26.7 %) | | 7 cm | | | |
| | | | 8 cm | 4 (6.9 %) | 3 (5.2 %) | 6 (10.3 %) |
| 10 cm | | 1 (6.7 %) | 10 cm | 3 (5.2 %) | 1 (1.7 %) | 3 (5.2 %) |
| | | | 12 cm | 2 (3.4 %) | | 4 (6.9 %) |
| | | | 14 cm | 1 (1.7 %) | | |
| | | | 18 cm | | | 1 (1.7 %) |
| | | | 20 cm | | 1 (1.7 %) | |

Table 10 compares the depths at which 90 % of the total Cs-137 concentration in soil samples was measured (not migration head depths). The results on the left are from a research paper published following the CNPP accident (Ivanov et al., 1997). The results on the right are from soil samples collected at the present study site. The numbers of samples and the years after each respective accident differ between the two projects. However, the current study shows that Cs-137 migrated downward more quickly in the region affected by the FDNPP accident than in the region affected by the CNPP accident. The faster





downward migration at the present study site may be attributed to the site's low bulk density of 0.44 g cm$^{-3}$ and to its sandy soils.


According to Fig. 23, the models performed well for the mid-depths, around 6.0 to 20.0 cm, which is important for mitigation purposes because the average migration head depth of the FDNPP-derived Cs-137 was 20.79 cm. The reasons why the model performance was poor in the depths of 2.0 to 6.0 cm (Fig. 17, Fig. 20, and Fig. 23) need further analysis, with more detailed soil property and the temporal change data of Cs-137 in the forest. The current assumption is that physical or biological factors

that affect Cs-137 accumulation in those depths were not included in the models, such as the effects of Cs-137 migration from surface litter and vegetation, organic matter in the soils, or infiltration patterns (see Sect. 7.4 below).

### 7.3 The effects of DEM resolutions

Overall, the topographic parameters extracted from the 10 m DEM returned higher explanation percentages than those extracted from the 1 m DEM (Tables 6 and 8). The results contradicted the authors' initial assumption that the 1 m DEM would

capture the topography of the study site with microtopography. The descriptive statistics (Fig. 16) of the parameters showed that TWI displayed the largest difference between the 1 m DEM and 10 m DEM. The median TWI with the 1 m DEM was 1.63 while, with the 10 m DEM, the corresponding TWI was 5.45. The parameter with the second-largest difference was upslope distance (16.90 m with the 1 m DEM and 68.59 m with the 10 m DEM).

Upslope distance appeared in the best-performing parameter combinations for the 10 m DEM. The range of absolute plan curvature values narrowed with the 10 m DEM, although the COV increased among the measurements (Fig. 16). Heimsath et al. (1999) have found that curvature becomes scale-dependent in grids larger than 5 m, supporting the reduced range of plan curvature values with the 10 m DEM in this study.

Table 11 lists each topographic parameter's mean value ratios between the DEMs and with which DEM those parameters

appeared in the better-performing models. These findings led the authors to two new hypotheses: (1) topographic features appearing in the better-performing models with the 10 m DEM (slope degrees, upslope distance) had more effects on Cs-137 accumulation in a larger spatial scale than those parameters in the better-performing models with the 1 m DEM and (2) plan curvature and TWI (as already stated) reflected the physical processes in a spatial extent of less than 10 m.




**Table 7. Difference ratios of DEM resolutions and the mean measurements of topographic parameters.**

| Difference ratio | 1 m DEM | 10 m DEM | The DEM where each parameter appeared in the better-performing models |
|---|---|---|---|
| elevation | 1.00 | 1.01 | 1 m / 10 m DEM |
| slope degrees | 1.08 | 1.00 | 10 m DEM |
| upslope distance | 1.00 | 4.05 | 10 m DEM |
| plan curvature | 1.00 | 100.00 | 1 m DEM |
| TWI | 1.00 | 3.15 | 1 m DEM |

**7.4 Soil Properties**

Soil properties (water content and bulk density) explained Cs-137 values in Bq kg$^{-1}$ units at significant levels (p-values of 0.05
for water content and of 0.06 for bulk density) (Table 7), confirming that Cs-137 accumulation in Bq kg$^{-1}$ depends on soil
texture and water content. For Cs-137 values in Bq m$^{-2}$ units, the prediction performance with soil properties was less than Bq
kg$^{-1}$ units.

As mentioned in Sect. 7.2, the explanation percentages of models were higher in the mid-depths. Separate Pearson's correlation
index analyses demonstrated that water content and Cs-137 values were correlated at depths between 6.0 and 22.5 cm (index
average of 0.65 for Cs-137 in Bq kg$^{-1}$; 0.60 for Cs-137 in Bq m$^{-2}$). The correlation analyses also found that TWI was weakly
correlated with Cs-137 at the depths (index average of 0.44 for Cs-137 in Bq kg$^{-1}$; 0.38 for Cs-137 in Bq m$^{-2}$). A hypothesis in
this regard is that water content, resulting from infiltration and surface water pooling, has connections with Cs-137 levels and
helped GAM models predict Cs-137 concentrations in the mid-depths despite the large COVs of Cs-137 values at the same
depths (Fig. 11–Fig. 12).

**7.5 The Significance of vegetation cover types and hillslope aspects**

The Tukey's HSD tests (Table 9) revealed the inter-effects of topography and vegetation cover types in predicting Cs-137
concentrations. When the samples were separated by evergreen and deciduous areas, the parameter pairs containing upslope
distance did make mean differences in the predictions against either elevation or slope degrees. At this particular study site,
what matters for Cs-137 predictions was the locations of samples relative to the ridges – not a linear distance from the hill
bottom (the elevation) – and the effects were enhanced by the vegetation growing on the slope. Hillslope aspects did not show
significant differences in the predictions at this study site.



### 7.6 Model extrapolation performance

This study attempted a numerical spatial extrapolation using a model with only topographic parameters. The GAM model used for extrapolation ("elevation + slope degrees + upslope distance") had a deviance explanation percentage of less than 40 %,

and the model extrapolation result was 12 % less than the Japanese government's estimate; in other words, the model predicted up to 82 % of the expected total Cs-137 budget. The model needs to be improved by testing other topographic parameters and adding samples, if possible, in the future. Still, the extrapolation result was better than the numerical model suggested, and the result indicates the robustness of topographic parameters in geomorphological predictions.

### 8 Conclusions

This study examined and attempted to predict Cs-137 concentrations in a forest using a numerical approach. The study focused on the effects of topography on Cs-137 concentration levels and compared the effects of the soil properties. The results showed that combinations of topographic parameters affected prediction performance, and combining multiple parameters returned better predictability. The comparison of the topographic parameters extracted from DEMs with different spatial resolutions showed the variabilities of parameter ranges, depending on the parameter extraction methods. Soil property data can enhance

the performance of topography-based models for Cs-137 concentration predictions. However, the vertical results showed that topography and soil properties do not explain Cs-137 concentration consistently throughout the sampled soil depths. Although their relative prediction power was in sync in individual depth intervals, the degrees of explanation percentages themselves differed among the layers. Further analysis using mean differences in Cs-137 predictions revealed the inter-effects of topography – particularly upslope distance at this study site – and vegetation cover types in predicting Cs-137 concentrations.

Lastly, the vertical migration speed of Cs-137 in the Fukushima forest was faster than the corresponding speed in the area affected by the CNPP accident. The final extrapolation attempt demonstrated the possibility of predicting radionuclide contamination based on topographic features.

### 9 Limitations

#### 9.1 Comments on temporal dynamics

In this study, Cs-137 measurement began in 2016, and the physical translocation of Cs-137 prior to 2016 was not considered. The physical processes that might have affected Cs-137 movements before 2016 is a black box; prior to 2016, no soil monitoring was conducted in the study-site forests. The only data the authors had were the initial Cs-137 deposition estimate for the study site region, released in March 2011 by the Japanese government and the United States Department of Defense (MEXT, 2011). Cs-137 values are decay-normalized in this article, and thus, the research does not consider the physical

translocation of Cs-137 over the three-year sampling period. Channel discharge Cs-137 data, which would have provided a clue on surface Cs-137 loss at the study site over the years, also does not exist. Considering that the average soil bulk density



was 0.44g/cm³, which is a very low dry bulk density, the extent of Cs-137 loss due to surface runoff was considered to be limited (Bharati et al., 2002; Meek et al., 1992). However, the effects of constant precipitation in rainy seasons, as well as the intense typhoon season precipitation in Japan, cannot be ignored. Still, all soil sampling campaigns were conducted at
approximately the same time of year during all three data collection years – after the end of the rainy season in Japan. All sampling campaigns avoided intense precipitation immediately prior to sample collections. Thus, the ground and soil conditions did not differ significantly.

The authors conducted a separate temporal investigation of changes between the 2016 and 2018 samples. This investigation showed that the average COG depth among samples moved down less than 1.0 cm and the average migration head depth
moved up 1.75 cm between the two years. Although there were variations among the samples, the overall Cs-137 vertical accumulation patterns in the soils did not change drastically over two years. These temporal changes will be addressed in a separate article.

### 9.2 Other Limitations

1) The results of this research apply only to the study site, but the analytical methods can be applied to other sites.
Whether the same topographic factors significantly affect Cs-137 must be verified in other locations.
2) The importance of organic matter in understanding Cs-137 behavior has been recognized (Claverie et al., 2019; Doerr and Münnich, 1989; Dumat et al., 1997; Dumat et al., 2000; Fan et al., 2014; Korobova et al., 2016; Mabit et al., 2008; Staunton et al., 2002; Takenaka et al., 1998; Tatsuno et al., 2020), and carbon/nitrogen (C/N) ratio measurement for the collected samples was attempted in 2016. However, due to time constraints and equipment issues, data were
extracted for only a small portion of the samples. Research using soils from the same forest (Tatsuno et al., 2020) has reported the effects of dissolved organic matter on Cs transport. The authors, therefore, refer readers interested in the topic to Tatsuno et al. (2020).

### Author contributions

Yasumiishi, M. conceived and designed the project; Yasumiishi, M. and Nishimura, T. conducted field data collections and
laboratory analysis; Yasumiishi, M. conducted data analysis and graphing; Nishimura, T., Aldstadt, J., and Bennett, SJ. Provided critical feedback on data analysis; Yasumiishi, M. drafted the manuscript; Bittner, T. advised on data concepts. All authors provided critical feedback to finalize the manuscript.

### Competing interests

The authors declare that they have no conflict of interest.



**Acknowledgments**

This study was supported by the following awards: The NSF East Asia and Pacific Summer Institutes (EAPSI) 2016 program [award number: 1614049], the NSF Doctoral Dissertation Research Improvement Award - Geography and Spatial Sciences Program (DDRI-GSS) [award number: 81809], and the College of Arts and Sciences Dissertation Fellowship of the University at Buffalo. Field sampling, soil sample lab tests, and radioactivity measurements were accomplished with the support of the Laboratory of Soil Physics and Soil Hydrology, Graduate School of Agricultural and Life Sciences, the University of Tokyo, Japan. Mr. Kinichi Okubo, a farmer in Iitate Village, kindly provided his land for this research.

**Appendices**

**Appendix A. The standard setting for GAM models.**

Model ← gam(Cs-137 ~ s(x₁) + s(x₂) + s(x₃), data=data, method="REML", bs="cr", family=Gamma(link=log)),

where s indicates that the spline-based smooth term is applied to the variable; bs sets the (penalized) smoothing basis; REML indicates restricted maximum likelihood; and cr indicates that cubic splines are used. This GAM setting was consistent for all model runs.

**Appendix B. The GAM model setting with a categorical term.**

Model ← gam(Cs-137 ~ Vegetation Type/Location + s(x₁) + s(x₁, by=Vegetation Type/Location, m=1) + s(x₂) + s(x₃), data=data, method="REML", bs="cr", family=gamma (link=identity))

The x₁ variable was evaluated by itself and then with the subcategories. The "m=1" term ensured that the model put the penalty on the first derivative for the difference smooths. The "by=" term was applied to each variable in the GAM equation, one at a time.

**Appendix C. Cs-137 decay normalization equation.**

The Cs-137 decay constant is calculated as follows (Eq. A1):

$$K = \frac{\ln 2}{T_{\frac{1}{2}}}$$ (A1),

where $K$ is the decay constant and $T_{1/2}$ is the half-life. The half-life of Cs-137 is 30.17 years; thus, $K = 0.022975$ year⁻¹. The radioactive decay formula is as follows (Eq. A2):



$$No = \frac{Nt}{e-kt}$$ (A2),

where $No$ is the original Cs-137 value, $Nt$ is the Cs-137 value after time $t$, and $k$ is the decay constant (IAEA-TECDOC, 2003).

**Appendix D. Generalized cross-validation (GCV) equation.**

$$V_g = \frac{\sum_{i=1}^{n} n(y_i - \hat{f}_i)^2}{[n - tr(A)]^2} \text{ (Wood, 2017)}$$ (A3),

where $V_g$ is the generalized cross-validation score, $y_i$ is the excluded data, $\hat{f}$ is the estimate from fitting to all the data, and $tr(A)$ is the mean of the model matrix $A_{ii}$.

**Appendix E. Akaike information criteria (AIC) equation.**

$$AIC = -2l(\hat{\theta}) + 2p \text{ (Wood, 2017)}$$ (A4),

where $l(\hat{\theta})$ is log-likelihood and $p$ is the number of identifiable model parameters (usually, the dimension of $\theta$).



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
