# Peer review of "Assessing the Effect of Topography on Cs-137 Concentrations within Forested Soils due to the Fukushima Daiichi Nuclear Power Plant Accident, Japan"

_Earth Surface Dynamics, 2020_

## Referee Comment (RC1) · Anonymous Referee #1 · 18 Nov 2020

Dear Authors and Editors!

I with great pleasure read manuscript entitled "The effects of topography and soil properties on radiocesium concentrations in forest soils in Fukushima, Japan". At first, please let me apologies for slow reply and postponing in review process. I had a seasonal flu. The Fukushima Dai-ichi nuclear accident lead to a great contamination by radioisotopes of a large area at Honshu island (Japan). The article describes a distribution of Cs-137 in forest soils at Fukushima prefecture. Such data is useful for current monitoring and forecasting of radiocesium concentration in study area. The paper fits to the "Earth Surface Dynamics" journal scope, but I have some suggestions and com-

ments. Please see them below. 1. In Abstract, 1 sentence. When the soil samples were collected? Better to add the years. As I understand from text, they are: 2016, 2017 and 2018. 2. 3 line of Abstract. Instead "accumulation" better "distribution". 3. LL 45. "Once released, Cs-137 takes two pathways in the environment. It may be dissolved in water (Iwagami et al., 2015; Osawa et al., 2018; Sakuma et al., 2018; Tsuji et al., 2016) or adsorbed into soil particles". I suggest change to "Once released, Cs-137 may be dissolved in water (Iwagami et al., 2015; Osawa et al., 2018; Sakuma et al., 2018; Tsuji et al., 2016) or adsorbed into soil particles". 4. Section 2.1. In my opinion, at the Site description the additional information is required and need some rewriting. In the beginning, better to show clearly which type of trees are covered the study area (in present version is could be found only in the middle part of section) and their Latin names, their average height and density; which types of soil are presented in study site; the angles of slopes (their range and values), as well m.a.s.l. 5. LL. 120. I guess "however, the FDNPP is not visible from this ridge" could be avoid. The FDNPP difficult to see from 35 km. 6. Fig. 5c. On some of the contour lines need to add the values. Also if possible - the Bergstrich lines. Much better if the maps are shown with a coordinate grid or put in the corner of maps the coordinates. 7. LL. 175. ". . .dried in an oven for about 24 hours at 105°C." I hope that this time were enough for drying. You could add that the samples were dried till constant weight. 8. LL. 195. "In the top layer, the average water content percentage was above 100 % because some samples were very moist." Probably is a mistake. For example, the field capacity of soil is average 30-50% and depends of soil texture. The field capacity is a moisture of "wet" soil, 2-3 days after rain or irrigation. 9. Table 4. In the name of table ". . .in this article" could be avoid. The sentence could be change to "Soil properties of the studied samples by depths." 10. Table 4. Average water content (%). Does the values are correct? 1.22% is very dry for "wet" soil. It is near to hygroscopic moisture of soil.

The parts of article with statistical analyses and using of models seems excellent. I not specialist in modeling thus, I recommend to contact with researchers who are working in such topic. I found some articles, which are very close: - Uncertainty assessment

method for the Cs-137 fallout inventory and penetration depth. - Modelling the extent of Cs-137 soil contamination patterns at the Kostica River basin (Bryansk Region, Russia). - Detailed study of post-Chernobyl Cs-137 redistribution in the soils of a small agricultural catchment (Tula region, Russia) It will be very nice if they (Drs. Papadakos, Linnik, Zhidkin) could give some recommendations on your paper.

Wish good luck in your present and future research work.

---

## Referee Comment (RC2) · Anonymous Referee #2 · 12 Dec 2020

Results and discussion are interesting. But, motivation of this study is poor. You reviewed some papers and showed table 1 about soil sampling papers. But you should review papers about topography effects. For example, Atarashi-Andoh et al (2015) JER explained topographic effects to variation of radiocesium. You need to do more carefully review and lead to motivation of this study from review results. Then you should compare your results and reviewed results in discussion section. If you doing that, this paper will be good.

Table 6 in p39 is Table 10? Table 7 in p41 is Tbale 11?

[Figure]

2020.

**ESurfD**

Interactive
comment

---

## Referee Comment (RC3) · Elena Korobova (Referee) · 17 Dec 2020

[referee-annotated manuscript omitted]

---

## Author Response (AR1)

Dear Editors/Reviewers,

The below are the comments from the reviewers and my responses.

The order is as follows:

Reviewer #1 comment → my response
Reviewer #2 comment → my response
Reviewer #3 comment → my responses (long)

You will notice that there were changes for model explanation percentages and Cs-137 values.

I informed the editors last year. While I was working on my second paper, after submitting the current paper, I noticed that the Cs-137 measurements were off for five samples. The Cs-137 values were low for the samples, and the values increased uniformly across the depths at the same rates.

It didn't change the paper structure, methodologies, or the overall results of the paper. It did make minor changes in descriptive statistics and the numbers in the tables.

The reason for the changes is this:

To get the Cs-137 concentration values (Bq/kg) for my samples, I went through a couple of steps.

1) The isotope lab, which I used, returned the raw 'count rates' (the total counts of Cs-134 + Cs-137).
2) Using formulas that the isotope lab provided, the count rates were converted to activity rates (Bq/kg). Then Cs-134 vs. Cs-137 ratio on that measurement date was calculated based on the decay rate and the number of dates since the accident.
3) Then, the Cs-137 values were normalized to a set date.

I didn't realize the error earlier because I was using only Cs-137 for this current paper. When I started working on the second paper, in which I use both Cs-134 and Cs-137, I noticed that the ratio was off.

I just wanted to make the reason for the changes.

[Figure]

I with great pleasure read manuscript entitled "The effects of topography and soil properties on radiocesium concentrations in forest soils in Fukushima, Japan". At first, please let me apologies for slow reply and postponing in review process. I had a seasonal flu. The Fukushima Dai-ichi nuclear accident lead to a great contamination by radioisotopes of a large area at Honshu island (Japan). The article describes a distribution of Cs-137 in forest soils at Fukushima prefecture. Such data is useful for current monitoring and forecasting of radiocesium concentration in study area. The paper fits to the "Earth Surface Dynamics" journal scope, but I have some suggestions and comments. Please see them below. 1. In Abstract, 1 sentence. When the soil samples were collected? Better to add the years. As I understand from text, they are: 2016, 2017 and 2018. 2. 3 line of Abstract. Instead "accumulation" better "distribution". 3. LL. 45. "Once released, Cs-137 takes two pathways in the environment. It may be dissolved in water (Iwagami et al., 2015; Osawa et al., 2018; Sakuma et al., 2018; Tsuji et al., 2016) or adsorbed into soil particles". I suggest change to "Once released, Cs-137 may be dissolved in water (Iwagami et al., 2015; Osawa et al., 2018; Sakuma et al., 2018; Tsuji et al., 2016) or adsorbed into soil particles". 4. Section 2.1. In my opinion, at the Site description the additional information is required and need some rewriting. In the beginning, better to show clearly which type of trees are covered the study area (in present version is could be found only in the middle part of section) and their Latin names, their average height and density; which types of soil are presented in study site; the angles of slopes (their range and values), as well m.a.s.l. 5. LL. 120. I guess "however, the FDNPP is not visible from this ridge" could be avoid. The FDNPP difficult to see from 35 km. 6. Fig. 5c. On some of the contour lines need to add the values. Also if possible - the Bergstrich lines. Much better if the maps are shown with a coordinate grid or put in the corner of maps the coordinates. 7. LL. 175. "...dried in an oven for about 24 hours at 105°C." I hope that this time were enough for drying. You could add that the samples were dried till constant weight. 8. LL. 195. "In the top layer, the average water content percentage was above 100 % because some samples were very moist." Probably is a mistake. For example, the field capacity of soil is average 30-50% and depends of soil texture. The field capacity is a moisture of "wet" soil, 2-3 days after rain or irrigation. 9. Table 4. In the name of table "...in this article" could be avoid. The sentence could be change to "Soil properties of the studied samples by depths." 10. Table 4. Average water content (%). Does the values are correct? 1.22% is very dry for "wet" soil. It is near to hygroscopic moisture of soil.

The parts of article with statistical analyses and using of models seems excellent. I not specialist in modeling thus, I recommend to contact with researchers who are working in such topic. I found some articles, which are very close: - Uncertainty assessment

method for the Cs-137 fallout inventory and penetration depth. - Modelling the extent of Cs-137 soil contamination patterns at the Kostica River basin (Bryansk Region, Russia). - Detailed study of post-Chernobyl Cs-137 redistribution in the soils of a small agricultural catchment (Tula region, Russia) It will be very nice if they (Drs. Papadakos, Linnik, Zhidkin) could give some recommendations on your paper.

Wish good luck in your present and future research work.

———————————————

Earth Surf. Dynam. Discuss.,
https://doi.org/10.5194/esurf-2020-72-AC1, 2020

[Figure]

Dear Referee #1 First of all, thank you so much for providing us your valuable comments. I will answer your comments below. The final manuscript will be revised, reflecting these answers.

1. In Abstract, 1 sentence. When the soil samples were collected? Better to add the years. As I understand from text, they are: 2016, 2017 and 2018.

Answer: Yes, that is correct. I will add the years to the abstract.

2. 3 line of Abstract. Instead "accumulation" better "distribution".

Answer: I will revise it.

3. LL 45. "Once released, Cs-137 takes two pathways in the environment. It may be dissolved in water (Iwagami et al., 2015; Osawa et al., 2018; Sakuma et al., 2018; Tsuji et al., 2016) or adsorbed into soil particles". I suggest change to "Once released, Cs-137 may be dissolved in water (Iwagami et al., 2015; Osawa et al., 2018; Sakuma et al., 2018; Tsuji et al., 2016) or adsorbed into soil particles".

Answer. I understand that you thought that 'takes two pathways' was too definite. I will revise the expression.

4. Section 2.1. In my opinion, at the Site description the additional information is required and need some rewriting. In the beginning, better to show clearly which type of trees are covered the study area (in present version is could be found only in the middle part of section) and their Latin names, their average height and density; which types of soil are presented in study site; the angles of slopes (their range and values), as well m.a.s.l.

Answer: Thank you for pointing out. I will provide additional information as much as possible. With regard to the trees, below is an excerpt from my dissertation draft. I will add forestry information from this part as well as additional data based on my observation and public record.

"These forests are not all 'natural' or 'native.' The Japanese government began tree-planting and land management projects in the late 17th century to mitigate the over-harvesting of lumber and land degradation due to the country's increasing population. Two types of trees that the government recommended for planting in northern Japan were cedar and Japanese cypress (Ministry of Agriculture Forestry and Fisheries, 2013). In the study site's forests, a few wooden sticks indicate when the most recent planting projects were completed and the number of trees planted. On the longest slope, where most of this study's samples were collected, 6,300 cypress saplings were planted in May 1998, and the planting area was 1.81 hectares (18100 m2). On the

hills on the east side, 74,100 red pine saplings were planted in April 1965, and the planting area was 14.83 hectors (148300 m2). In the region, there has been a history of charcoal production from the forests. According to a local farmer, the area was once cleared as pasture, and the forest hills were used as grazing ground for cattle. Cattle are still raised in the lowlands in this region. Whether the land-use history of the forests affects radionuclide behavior is an important question. However, since the radionuclide fallout happened recently, and no major forestry work (e.g., new planting) has been conducted since the accident, land-use history is not considered in the subsequent analysis. It would be interesting to compare radionuclide behaviors between 'natural' and 'planted' forests in future research."

5. LL. 120. I guess "however, the FDNPP is not visible from this ridge" could be avoid. The FDNPP difficult to see from 35 km.

Answer: I will delete the sentence. My intention was to demonstrate that there are natural obstructs (mountain ridges) between the plant and the study site. But I agree that the sentence was unnecessary.

6. Fig. 5c. On some of the contour lines need to add the values. Also if possible - the Bergstrich lines. Much better if the maps are shown with a coordinate grid or put in the corner of maps the coordinates.

Answer: A good point about the contour line values. I will add them. Regarding the coordinates, I intentionally did not add them. A farmer personally owns the study site. As you see in the manuscript, the contamination levels are high in the plot, and I do not want the farmer to suffer negative consequences because I disclose the exact location. If someone tries to find out the location, it will not be difficult (and some people know where the site is). But I would like to refrain from printing the exact coordinates.

7. LL. 175. ". . .dried in an oven for about 24 hours at 105_C." I hope that this time were enough for drying. You could add that the samples were dried till constant weight.

Answer: I did extended drying time for very wet samples. I will provide additional statements about drying procedure.

8. LL. 195. "In the top layer, the average water content percentage was above 100 % because some samples were very moist." Probably is a mistake. For example, the field capacity of soil is average 30-50% and depends of soil texture. The field capacity is a moisture of "wet" soil, 2-3 days after rain or irrigation.

Answer: Thank you for pointing out. Maybe I should not have used the term, 'moist.' Those soils were collected in wet areas, so they did contain those percentages of water when I measured their weights before drying. To make it clear, I can change the sentence to 'because some samples were collected in wet areas and contained water exceeding field capacity. All samples were weighted as collected in the field before drying." I could have set aside those samples and let excess water evaporate for a few days. However, that would not reflect the natural condition. So I decided to go head and measure them.

9. Table 4. In the name of table ". . .in this article" could be avoid. The sentence could be change to "Soil properties of the studied samples by depths."

Answer: Thank you. I'll change the title.

10. Table 4. Average water content (%). Does the values are correct? 1.22% is very dry for "wet" soil. It is near to hygroscopic moisture of soil.

Answer: Thank you for pointing out!! I'll remove the decimals. Sorry, it was an error.

Earth Surf. Dynam. Discuss.,
https://doi.org/10.5194/esurf-2020-72-RC2, 2020

[Figure]

Results and discussion are interesting. But, motivation of this study is poor. You reviewed some papers and showed table 1 about soil sampling papers. But you should review papers about topography effects. For example, Atarashi-Andoh et al (2015) JER explained topographic effects to variation of radiocesium. You need to do more carefully review and lead to motivation of this study from review results. Then you should compare your results and reviewed results in discussion section. If you doing that, this paper will be good.

Table 6 in p39 is Table 10? Table 7 in p41 is Tbale 11?
* * *
Earth Surf. Dynam. Discuss.,
https://doi.org/10.5194/esurf-2020-72-AC2, 2020

[Figure]

Thank you so much for taking the time to read the manuscript and providing the feedback to us.

We understand that your point was that we did not explain sufficiently the reasons why we selected the topic, topography, in the first place. That is a good point. We will add past literature, including what you suggested, and the reasoning to the research motivation part.

Regards, Misa

Earth Surf. Dynam. Discuss.,
https://doi.org/10.5194/esurf-2020-72-RC3, 2020

[Figure]

The article is interesting as an attempt to carry out spatial estimate of Cs-137 distribution after the Fukushima accident in relation to topography of the contaminated basin. However, the applied sampling scheme does not allow to obtain results adequate to the selected scale of topograhical resolution (1 and 10 meters). Therefore I would like to recommend softening the presented conclusions.

Another comment concerns rather low values of the soil bulk density (ca 0.4-0.5 g/kg dry weight) and calculation of Cs-137 contamination density (Bq/m2 per soil layer) using Bq/kg DW data by recalculation on the basis of Bq/kg data after withdrawal of

rock fragments and roots from the soil core samples. The last operation may increase the real contamination density at the sampling point.

Please, see some other comments and recommendations in the attached file.

Please also note the supplement to this comment:
https://esurf.copernicus.org/preprints/esurf-2020-72/esurf-2020-72-RC3-supplement.pdf

Earth Surf. Dynam. Discuss.,
https://doi.org/10.5194/esurf-2020-72-AC3, 2020

[Figure]

Thank you for your thorough review and feedback. I will revise my manuscripts based your comments. Here, I reply to two of your comments.

1) Removal of roots and rocks: In the case of this study, soil samples were weighted after removing roots and rocks. It does make bulk density lower than the actual density it originally was. Looking back, soil weight and Cs-137 should have been measured twice before and after the removal. If I have an opportunity to repeat this study with

less time constraint, I will definitely try it. For this article, I will add comments about this issue. Also, I will make it clear that those removals where mostly done in the top layer.

2) Sandy soil with low bulk density: That was a surprise for us too. In shallow soils, the removal of roots and rocks must have affected the bulk density calculation. Overall, the soil itself was sandy, however, the soil structure in many samples were loose, not packed densely. As you suggested, I will add a couple of photos of actual samples.

The percentage of organic matter was not measured, due to time constraint. However, I do have data from a couple of samples (approximately 10% in the top 0-2 cm samples).

Referee #1 pointed out the water content % in the table seemed to be incorrect. I made an error in describing the unit. I will fix it.

Regards, Misa
* * *
[Figure]

[revised manuscript text omitted]

---

## Author Response (AR2)

**Response to Editor Comments (indented paragraphs in bold)**
Comments to the Author: esurf-2020-72, The effects of topography and soil properties on radiocesium concentrations in forest soils in Fukushima, Japan

General response from the authors
We greatly appreciate the detailed review and comments provided and the patience exhibited by the editorial team. With these detailed comments and suggestions, we undertook a major rewrite of the paper.

Major changes include the following:
1. The title of the paper has been revised to more accurately represent the material presented. It is now entitled "Assessing the Effect of Topography on Cs-137 Concentrations within Forested Soils due to the Fukushima Daiichi Nuclear Power Plant Accident, Japan."
2. The Introduction now includes information about the nuclear accident, a stronger justification for the current study, and clearly defined objectives for the current paper that are strongly aligned with the results presented.
3. The Methods section has been rewritten. All methodological information on sampling, soil and radiation analyses, DEMs, and statistical analyses have been consolidated into a single section and presented in a logical order.
4. All primary data collected within this study have been included in an Appendix.
5. Several plots and figures were removed, revised, and consolidated, and unnecessary analyses and discussions were deleted. The total number of figures were reduced from 26 to 13.

In the sections below, the comments from the referee and editor are shown in italics, and the authors' responses are shown in normal text and indented.

*This manuscript received three reviews and I have read it in detail myself. Two reviewers were broadly supportive, but reviewer two was sceptical, stating that the motivation of the study was poor. Reviewer one, was positive, though makes a number of important points, and doesn't comment explicitly on the statistical methods. Reviewer 3 comments critically about the vertical resolution in relation to the samples collected.*

*Whilst this is undoubtedly an interesting problem and data set I have to agree with reviewer two that the motivation for the study would need to be much more clearly conveyed if this is to be published. An interesting data set and statistical analysis is not sufficient for publication. The bar is high in terms of how this needs to be presented if this is to be understood by the broad audience of ESURFD. In the authors response, I don't think they addressed sufficiently the comments of R2.*

> We now have rewritten the Introduction to include more background information on nuclear accidents, a stronger justification for the overall goal of the research that focuses on the effects of topography on Cs-137 concentrations, and clearly defined objectives of the current paper that are in complete alignment with the results presented. This scientific motivation and research goals also have been included in the Abstract and Conclusions, where appropriate.

*It is not reasonable to assume that readers will be familiar with all the statistical methods as well as the radiocesium literature.*

We have provided a more complete description of the statistical models employed, and we have expanded the presentation and discussion of model results and their application herein.  We then connected the model results to the overall goal of the paper, which is the effect of topography on Cs-137 concentrations.

*In summary, this manuscript takes a statistical approach to determine whether geomorphological parameters influence the accumulation of radiocesium following a major nuclear disaster. A number of samples were collected over a 3 year period following the disaster from a relatively restricted area, over a relatively narrow range in altitude. The radiocesium was determined and a number of statistical models are used to test whether there are links with key topographic features such as elevation, slope, curvature and vegetation type. It is never really laid out to the reader why 137Cs might correlate with such topographic parameters (the list on pp17 is the best but it should have come much sooner). What are the underlying physical/chemical/biological mechanisms (they get lost in the text and should be stated right at the beginning). On the one hand altitude might relate to the original deposition of 137Cs (orographic effect), and vegetation type (and elevation) might relation to physical/biological/chemical processing of 137C post deposition. Why should slope and curvature matter (this is briefly stated but not elaborated)? If these are the key parameters that control the accumulation (retention) of 137Cs in the environment then some clear figures with the data are required to show this both in an individual and multivariate manner. The current figures do not do a sufficient job of conveying a clear message from the data and are set out in a disorganised manner. There are too many figures for the message that is being conveyed. There would be clearer, simpler more concise and convincing methods to elucidate a complex yet fascinating dataset.*

We now provide a literature review and a theoretical framework for the topographic indices examined here and their potential impact on Cs-137 concentrations in soils at and just below ground surface.  We rewrote the Introduction and Methods sections to provide this important context.  In addition, the number of figures has been reduced significantly, and the primary data collected have been included as an Appendix in an effort to present a clearer and more concise reporting of the research.

*There are a series of major issues that would need to be addressed if this were to be considered further for publication. Based on the author response to the initial reviews I'm not convinced that such a whole-scale rewrite is possible. There are organisational, methodological and grammatical issues which must all be addressed. As is, this does not read like a polished paper, but more like a detailed thesis or specialist report. Unfortunately, in the version I read, the figure numbering appears to have gone wrong which doesn't help. It may be, that it would be more appropriate in a specialised journal.*

The paper has undergone a complete rewrite.  While the results remain the same, we now provide much greater justification for the work, clearer separation of methods and results, expanded discussion and context for the methods employed, and a reduced number of figures to better focus the presentation and interpretation of the results.

*Specific comments (not exhaustive)*
*1. Generalized additive models (GAM)*
*The statistical models (for example the Generalized additive models (GAM)) seem complex for what the data shows. The GAM method is not well explained. What exactly is being minimised to generate the residuals? I'm not sure the text and figures do justice to the data? A large number of the figures (and there are too many) show deviance or residuals. It is difficult to ascertain from these figures the validity*

*of the proposed covariation or not. I don't think it is sufficient to say that the "gam.check ()" function was used. If readers aren't familiar with R or this particular function in R, this will convey very little.*

> We now include a greater explanation of GAMs, their mathematical basis, the approach used to create the models, and a description of the indices used to assess model performance. The phrase "deviance explained percentages" is used, which is the generalization of $R^2$ in the GAM algorithm. We also explained the gam.check() function employed in the R-package.

*There is no obvious or particularly intuitively relationships between 137-Cs and topographic parameters. For example, in section 6.2 accumulation patterns on a simple representative hillslope, 3 figures are shown. The first appears to show the relationship between elevation and 137-Cs. It is clear that there is no simple relationship. Perhaps this is why a complex GAM is required. What is the trend line on this diagram? What is the simple representative hill slope? The second figure appears to show slope vs elevation, but how does this relate to 137Cs?*

> It is precisely because there is no obvious relationship between Cs-137 concentration and topographic indices that we explored statistical models. GAM is a statistical detection tool to assess quantitatively relationships or trends amongst parameters that other regression methods fail to address. The graphs noted by the referee have been updated and simplified to improve the communication of the results and their interpretation.

*Perhaps clearly subheadings should guide the reader through the data analysis in a more systematic way such as:*
*I) Elevation*
*II) slope*
*III)soil water content*
*IIII) density*

> In our rewrite, we have revised all headings and reorganized all results and discussions to greatly improve the logical flow of the paper. In the current version, we now discuss the influence of each topographic and soil property parameter on Cs-137 concentrations. This section is intended to provide a clearer and stronger interpretation of the GAM results and the impact of topography on Cs-137 concentrations.

*At the moment, the data description and discussion appear to be driven by the statistics (four tests), rather than clearly testing a hypothesis with a clear physical or geomorphological basis. If the 4 tests were retained, they should be laid out much more clearly so the reader can follow. An example of how I was unable to follow even the Single parameter GAM results is given below:*

> We have eliminated the Tukey test (which tested the significance of vegetation cover types and slope aspects) and the spatial prediction aspects of the paper, so that our current effort remains entirely focused on topography. While we recognize that vegetation may play an important role in modulating Cs-137 concentrations and that spatial predictive analytics could be of interest, these previous discussions did not bring added value to the paper. In addition, we now provide all primary data collected in this study in an Appendix, to further inform the reader.

*Page 28:*
*As I understand section 6.2, it is examining individual parameters as a controlling factor on 137-Cs. It is a*

*pity tat it is not possible to indicate the lack of correlation (as I understand it) with a figure. However, whatI really don't understand is the following sentence: "None of the single parameters in either table returned deviance-explained percentages above 36 % (which was found via a model including water content)."*
*I don't understand how the water content is being modelled at this point. How the single parameter GAM works has not been well enough explained. Being single parameter, I had clearly wrongly assumed that it meant it was dealing with one parameter at a time.*

> For simplicity and clarity, we presented the GAM results in a table rather than graphically, and we also sought to improve the structure and context of the supporting discussion.  The sentence in question has been revised accordingly.

*2. Section 4.1 and 4.2: DEM work*
*This section seems to deal with extracting key parameters from the different resolution DEMS. However, it also seems to mix in 137Cs measurements (line 281). This doesn't seem like the right place to know the units of 137Cs. The work on the DEM resolutions is clearly important, but equally it is a distraction from the main message of the manuscript. Can section 6.3.1 move to an appendix? Equally, the organisation seems to be confused. Later on in the manuscript, Fig 15 (Slope degrees of sampling points along elevation (m)) seems to be entirely related to the DEM work and have nothing to do with radiocesium. There are clearly major organisational problems.*

> We restructured the article in the following ways.  All methodological information regarding the DEMs and the topographic indices are now presented in the Methods section, which also includes additional information for the selection of these indices in the context of landscape processes.  We include all primary data collected in this study, including the topographic indices extracted from the DEMs, now compiled as a table in an Appendix.  Lastly, we have reorganized the presentation of the results to strengthen the narrative and provide a logical sequence for the topics introduced.

*3. Figures*
*There seems to be a major problem with the numbering of the figures.*
*Figures should be understandable just by looking at the figure and the caption. For example, Fig 12 (pp36). What are the solid red lines. In the caption, it refers to top row graphics and bottom row graphics. What are these?*

> First, we reduced by one-half the total number of figures in the paper.  Second, we have revised many of the remaining figures and their captions to improve their graphical presentation and ease in interpretation.  Figure 12 noted here has been deleted.

*Fig 4: It is a shame that there is nothing higher resolution available than this. As is, it is not very satisfactory. Could this not be incorporated into fig 5, plotting the sample locations in 3 different colours (if I have understood the figures)?*

> We eliminated several figures, and now present a single image (Fig. 3) to demarcate the sampling locations clearly and efficiently.

*On Fig. 15, what are the red dots. It is not clear from the Fig or the caption.*

The red dots have been changed to numbers, which refers to the four samples with the highest Cs-137 concentrations.

*Line 432: "In Fig. 13, both trend lines of Cs-137 in Bq m-2 values show upward trend along elevation." Fig 13 appears to show depth vs 137Cs and COV.*

The revised figures (Figs. 7 and 8) now show Cs-137 distributions plotted against elevation.

*Figure 3. An aerial views of the study site, facing northwest.*
*This would be fine in a Phd thesis, but is not appropriate in a paper. In addition, it does not appear to be an aerial view as is stated.*

This figure has been removed.

*Figure 6: The low water content appears to contradict the 100% statement (referred to below). There appear to be a large number of outliers on the high side. Would this be best shown as a histogram? It is hard to visualise how much data is in the box.*

We converted the tables to boxplots, now shown as Fig. 6.

*Figure 8: Nice to see a photo, but is it needed? What does the photo convey scientifically? This would be fine in a thesis, but not a paper.*

This figure has been removed.

*Can figures 9 and 10 be combined with fig 8 that comes after 9 and 10? Or combine 8, 12 and 13, the interesting depth figures.*

To improve paper clarity, we combined several plots into single figures.  Figs. 8 and 9 are now presented as Fig. 4, Figs. 10, 11, and 12 are now presented as Fig. 5, and Figs. 13 and 14 are now presented as Fig. 7.  We also modified these graphs to improve their interpretation by the reader.

*Fig.17: Hard to know what it means.*

This figure has been removed.

*Can Fig. 9, 10 and 20 be combined? (Pp30 and 31)*

Where possible, we have combined many plots into single figures, which was done to improve the clarity and presentation of the results.

*The Fig on pp35 appears to be very useful, comparing model predicted 137Cs with data. A pity there is no fig number or caption.*

As it happens, this figure has been removed.

*4. Writing style and conciseness:*
*There are numerous examples where the writing style does not have the conciseness required for a scientific publication. Avoid phrases like "This section displays" and make a point directly.*
*Other examples (non-exhaustive): "Literature review" Seems like an odd heading for a scientific paper. Sounds more like a thesis. "When driving into the study site region, visitors see hills and mountains covered with forests. Winding, narrow roads connect". Not appropriate for a concise scientific paper. "natural" or "native". Choose one option*

We carefully edited the paper to address any lapses in language and word usage. All changes were applied as suggested.

*Line 140: Land use comments: This is not the right place for this comment. This section is about the study site. If and use matters, then include it. If not, then say it doesn't*

Amended as suggested

*Line 157: perhaps change to "to confirm anomalies observed in previous years…"*

Amended as suggested

*Line 159: change to "but mostly on the south-west side due to accessibility"*

Amended as suggested

*Liner 190: Particles larger than 2mm were removed with a sieve*

Amended as suggested

*Line 198: The unit of measurement is the kiloelectron volt (KeV), equivalent to the kinetic energy gained by an electron falling through a 1 volt potential.*

Amended as suggested

*Line 215: "Approximately less than". Just less than*

Amended as suggested

*Line 217. Missing full stop and space.*

Amended as suggested

*Line 226: "Only some of the samples were tested for texture because of time and human resource constraints during the limited lengths of the first author's stays in Japan." Remove. Informal detail not needed.*

Amended as suggested

*Other*
*Line 433: Camelback. This is not scientific terminology. The manuscript goes to a great effort to produce a set of descriptive statistics to define relationships between parameters. This shouldn't be undermined by colloquial vocabulary.*

Amended as suggested

*Center of gravity (COG) depth (cm): What is this?*

This term has been deleted.

*"Migration head depths" are not defined. This appears to be first mentioned on line 413.*

We now clearly define this parameter, shown graphically in Fig. 7.

*Soil properties are discussed, but not in great detail. What about the cation exchange capacity of the soils for example that Cs has a very high affinity for. Presumably the cation exchange capacity is thought to be constant in these soils. Is this also true with depth. It seems to me that one major variable that has been overlooked in this analysis is the chemistry. How much of the Cs is partitioned between aqueous Cs and the soil?*

McHenry and Ritchie (1977) stated that CEC was an important factor to consider for Cs-137 modeling. There are reasons why the soil chemistry topic was not incorporated into this study.
1. We did not have publicly available soil chemistry data for the study site soils.
2. We did conduct pH, CEC, and C/N ratio tests on a very limited number of samples selected for texture analysis and other research purposes (not included in this article). But the total number of samples tested was relatively low due to a lack of time and resources. We do believe the data presented herein was sufficient for the purpose of the objectives of the current paper.
3. It should be noted that soil chemistry data might not explain the Cs-137 concentration patterns directly. For example, CEC becomes affected by the percentages of organic matter in the soils and CEC itself might not linearly explain Cs adsorption rate (Mensah, et al., 2020).
4. There is ample literature that has incorporated chemical properties with Cs-137 analysis (Matsunaga, et al., 2013, Mori, et al., 2015, Parajuli, et al., 2013, Takahashi, et al., 2015).

*How can the average water content>100%. This was also asked by reviewer 1. I don' understand the author response. How can the moisture be in excess of 100%. I must be mis-understanding the definition.*

It is now made clear that we report 'mass water content,' which can achieve values higher than 100%.

References for this note:
Matsunaga, T., Koarashi, J., Atarashi-Andoh, M., Nagao, S., Sato, T. and Nagai, H.: Comparison of the vertical distributions of Fukushima nuclear accident radiocesium in soil before and after the first rainy season, with physicochemical and mineralogical interpretations, Science of the total environment, 447, 301-314, doi:10.1016/j.scitotenv.2012.12.087, 2013.

McHenry, J. R. and Ritchie, J. C.: Physical and chemical parameters affecting transport of 137Cs in arid watersheds, Water Resources Research, 13, 6, 923-927, 1977.

Mensah, A. D., Terasaki, A., Aung, H. P., Toda, H., Suzuki, S., Tanaka, H., Onwona-Agyeman, S., Omari, R. A. and Bellingrath-Kimura, S. D.: Influence of Soil Characteristics and Land Use Type on Existing Fractions of Radioactive 137Cs in Fukushima Soils, Environments, 7, 2, 16, 2020.

Mori, K., Tada, K., Tawara, Y., Ohno, K., Asami, M., Kosaka, K. and Tosaka, H.: Integrated watershed modeling for simulation of spatiotemporal redistribution of post-fallout radionuclides: application in radiocesium fate and transport processes derived from the Fukushima accidents, Environmental Modelling & Software, 72, 126-146, 2015.

Parajuli, D., Tanaka, H., Hakuta, Y., Minami, K., Fukuda, S., Umeoka, K., Kamimura, R., Hayashi, Y., Ouchi, M. and Kawamoto, T.: Dealing with the aftermath of Fukushima Daiichi nuclear accident: decontamination of radioactive cesium enriched ash, Environmental science & technology, 47, 8, 3800-3806, 2013.

Takahashi, J., Tamura, K., Suda, T., Matsumura, R. and Onda, Y.: Vertical distribution and temporal changes of 137Cs in soil profiles under various land uses after the Fukushima Dai-ichi Nuclear Power Plant accident, Journal of environmental radioactivity, 139, 351-361, doi:10.1016/j.jenvrad.2014.07.004 2015.

---

## Author Response (AR3)

Author response to the referee.

The authors very much appreciated the suggested revisions offered by the referee. All proposed changes noted by the referee were amended as suggested.

The referee asked that we include thoughts as to why water content, bulk density, and elevation are important parameters for consideration. As noted herein, the statistical analyses employed are empirically-based approaches to derive predictive relationships amongst the available data. That is, the results are wholly dependent on the limited data collected. These analyses do not provide a mechanistic explanation as to why specific parameter combinations were shown to be more important. The discussion of these results sought to emphasize both the power and limitations of this approach while minimizing any discussion that would be considered speculative.

We made minor corrections. These changes did not affect the discussions or conclusions. We also added notes to Appendix B tables to make the nature of the data we presented clear.

Sincerely,

Authors

[revised manuscript text omitted]